

# Direct high-precision radon quantification for interpreting high frequency greenhouse gas measurements

Dafina Kikaj[1], Edward Chung[1], Alan D. Griffiths[2], Scott D. Chambers[2], Grant Forster[3,4], Angelina Wenger[5], Penelope Pickers[3,4], Chris Rennick[1], Simon O'Doherty[5], Joseph Pitt[5], Kieran Stanley[5], Dickon Young[5], Leigh S. Fleming[3,6], Karina Adcock[3], and Tim Arnold[1,7]

[1]National Physical Laboratory, Teddington, UK
[2]Australian Nuclear Science and Technology Organisation, Locked Bag 2001, Kirrawee DC NSW 2232, Australia
[3]Center for Ocean and Atmospheric Sciences, School of Environmental Sciences, University of East Anglia, Norwich, UK
[4]National Centre for Atmospheric Science, University of East Anglia, Norwich, UK
[5]School of Chemistry, University of Bristol, Bristol, UK
[6]Now at: GNS Science, Gracefield, Lower Hutt, 5040, New Zealand
[7]School of GeoSciences, University of Edinburgh, Edinburgh, UK

**Correspondence:** Dafina Kikaj (dafina.kikaj@npl.co.uk)

**Abstract.** We present a protocol to improve confidence in reported radon activity concentrations, facilitating direct site-to-site comparisons and integration with co-located greenhouse gas (GHG) measurements within a network of three independently managed observatories in the UK. Translating spot measurements of atmospheric GHG amount fractions into regional flux estimates ('top-down' analysis) is usually performed with atmospheric transport models (ATM), which calculate the sensitiv-

ity of regional emissions to changes in observed GHGs at a finite number of locations. However, the uncertainty of regional emissions is closely linked to ATM uncertainties. Radon, emitted naturally from the land surface, can be used as a tracer of atmospheric transport and mixing to independently evaluate the performance of such models. To accomplish this, the radon measurements need to have a comparable precision to the GHGs at the modelled temporal resolution. ANSTO dual-flow-loop two-filter radon detectors provide output every 30 minutes. The measurement precision at this temporal resolution depends on

the characterisation and removal of instrumental background, the calibration procedure, and response time correction. Consequently, unless these steps are standardised, measurement precision may differ between sites. Here we describe standardised approaches regarding 1) instrument maintenance, 2) quality control of the raw data stream, 3) determination and removal of the instrumental background, 4) calibration methods and 5) response time correction (by deconvolution). Furthermore, we assign uncertainties for each reported 30-minute radon estimate (assuming these steps have been followed), and validate the

final result through comparison of diurnal and sub-diurnal radon characteristics with co-located GHG measurements. While derived for a network of UK observatories, the proposed standardised protocol could be equally applied to two-filter dual-flow-loop radon observations across larger networks, such as the Integrated Carbon Observation System (ICOS) or the Global Atmosphere Watch (GAW) baseline network.





# 1 Introduction

The Paris Agreement aims to keep this century's mean global temperature rise well below 2 °C and pursue efforts to limit it to 1.5 °C above pre-industrial levels. Meeting this goal requires significant improvement in our understanding of GHG emissions to enable the most efficient mitigation policies to be implemented. Current GHG emission information is dominated by "bottom-up" (prior) estimates derived from aggregated activity data, emission factors and facility-level measurements (from local to country scale) that rely on reported data and knowledge of natural systems (Ciais et al., 2014; Gurney et al., 2016; Nisbet

and Weiss, 2010). Since there can be large uncertainties associated with the spatial and temporal variability of these emission factors it is prudent to seek independent verification of the resulting emission estimates. To this end, a distributed network of precise, high frequency, in situ GHG observations can provide opportunities for independent "top-down" verification of GHG emission estimates.

In recent decades many GHG monitoring stations have been developed throughout Europe. The Integrated Carbon Observa-

tion System (ICOS) (https://www.icos-cp.eu/, last access: 24 March 2024) constitutes Europe's primary research infrastructure for the provision of in situ standardized, traceable, high-precision observations of atmospheric GHG amount fractions. This atmospheric monitoring network includes 46 stations across 16 European countries (Yver Kwok et al., 2015; Yver-Kwok et al., 2021) (https://www.icos-cp.eu/observations/atmosphere/stations, last access: 25 March 2024). The corresponding UK atmospheric GHG network is UK DECC (UK Deriving Emissions related to Climate Change). Between 2012 and 2014 UK DECC

was expanded from one to five stations. Since 1987, the original station (Mace Head, MHD, on the west coast of Ireland) has been part of the World Meteorological Organisation (WMO) Global Atmosphere Watch (GAW) baseline network. The four more recent stations are: Tacolneston (TAC), Ridge Hill (RGL), Bilsdale (BSD), and Heathfield (HFD), each of which having at least one inlet 90 m or more above ground level (a.g.l.) (Stanley et al., 2018; Stavert et al., 2019).

The most commonly used top-down method for making GHG emission estimates is through the use of inverse modelling,

where high-quality GHG measurements are combined with atmospheric transport models (ATMs) and prior information to make optimal emission estimates (Arnold et al., 2018; Bergamaschi et al., 2015, 2018; Brown et al., 2023; Ganesan et al., 2015; Lunt et al., 2021; Manning et al., 2011, 2021). Another method is the Radon Tracer Method (RTM), which utilizes simultaneous, co-located observations of $^{222}$Rn (hereafter radon) and a GHG, in combination with an estimated radon source function. Different implementations of this approach allow either local or regional-scale GHG emission estimates to be made

from the same fetch region influencing the radon observations (Biraud et al., 2000; Grossi et al., 2018; Levin, 1987; Levin et al., 2021; van der Laan et al., 2009, 2014).

Despite recent improvements in the spatio-temporal density of observations, and excellent quality of in situ atmospheric GHG measurements (the relative uncertainties are often <0.1 %), as well as the mathematical elegance of ATMs (Baker et al., 2006; Tolk et al., 2008) large uncertainties remain in total annual top-down GHG emission estimates (e.g., ~10 % for the UK

$N_2O$ emission estimates; UK's National Inventory Report to the United Nations Framework Convection on Climate Change 2022). A key contributor to the overall uncertainty in top-down inverse model estimates of GHG emissions is the ATMs themselves. The challenge of quantifying ATM uncertainty is evident in tasks such as boundary layer height estimation and





parametrisation of meteorological variables, where errors are related to the resolution of the model (Tolk et al., 2008). As yet, no optimal method exists to evaluate the discrepancies between *a priori* and *a posteriori* emission estimates (bias). Until it is

possible to accurately quantify an ATM's uncertainty, full realization of the potential offered by high-quality atmospheric measurements from a comprehensive tower network cannot be achieved. This is a crucial step for improving national inventories and developing policy that the international community should have to hand while strengthening the 2015 Paris Agreement.

To this end, measurements of the naturally occurring, passive tracer radon could provide a means of evaluating ATM performance. Consequently, efforts to broaden the international radon monitoring network and harmonise the resulting measurements

not only stand to improve GHG emission estimates via atmospheric inversion studies, but also provide a second independent top-down method to estimate local- to regional-scale GHG emissions via the RTM.

Radon is the gaseous decay product of $^{226}$Ra (radium), a member of the $^{238}$U (uranium) decay chain, which is ubiquitous all over the Earth's crust. When emitted into the atmosphere, radon experiences the same atmospheric transport and mixing as all other gases released near surface. Being an inert gas, radon does not chemically react with any atmospheric constituents

and its low solubility makes it resistant to dry and wet deposition, leaving radioactive decay (half-life of 3.82 days) as its sole atmospheric sink. Its half-life, conveniently between the timescales of diurnal and synoptic atmospheric processes, is ideal for characterizing a wide range of meteorological phenomena (Galmarini, 2006; Kikaj et al., 2019; Williams et al., 2013; Zahorowski et al., 2004). It is therefore considered a powerful and convenient tracer at local, regional and global scales for improving, testing and validating ATMs (Dentener et al., 1999; Israël et al., 1966; Jacob et al., 1997; Taguchi et al., 2002;

Zhang et al., 2021).

Since its discovery in 1900, radon's unique physical characteristics have led to its use in studies of vertical mixing and air mass history (Eve, 1908; Wigand and Wenk, 1928; Wright and Smith, 1915). Early studies employed a variety of discrete measurement techniques but were often lacking in sensitivity or temporal resolution. Most significant progress in utilising radon as a relative tracer for vertical mixing and transport near the Earth's surface has been achieved since the 1960s, mainly driven

by advancements in continuous measurement techniques. Initially, public health (indoor) applications dominated instrument development, but the large indoor-to-outdoor gradient in radon activity concentrations (henceforth radon concentrations) limited the utility of these instruments in the outdoor environmental atmosphere. The first semi-continuous radon detector was developed in the mid-1960s (Taylor and Lucas, 1967). While suitable for near-surface inland measurements, this type of detector was not sufficiently sensitive for measurements at coastal or island sites. The capability to measure radon concentrations

typical of the remote marine atmosphere began to emerge post-1970 (Lambert et al., 1970; Polian et al., 1986; Pereira and Da Silva, 1989; Levin, 1987). More recently still, refinement of the original two-filter radon detector (Thomas and Leclare, 1970) by Whittlestone and Zahorowski (1998) greatly improved both the sensitivity and temporal resolution of radon measurements, better aligning them with advances in GHG measurements and modelling resolution. This growing collection of one- and two-filter radon monitors constituted the first of the "research grade" instruments. Following these developments, the

popularity of radon increased as a quantitative tracer in atmospheric modelling (Chevillard et al., 2002; Dentener et al., 1999; Jacob et al., 1997; Hirao et al., 2008; Mahowald et al., 1995; Zahorowski et al., 2004; Zhang et al., 2021) and estimation of



GHG fluxes on local to regional scale by the RTM (Biraud et al., 2000; Grossi et al., 2018; Levin, 1987; Levin et al., 2021; van der Laan et al., 2009, 2014).

Contemporary research-grade radon monitors are based on three fundamentally different measurement techniques: (i) in-
direct one filter $\alpha$- or $\beta$- activity detectors, which directly filter ambient aerosol-bound radon progeny from the atmosphere and count them, assuming equilibrium between atmospheric radon and its progeny (Biraud et al., 2000; Levin et al., 2002; Levin, 1987; Polian et al., 1986); (ii) direct two filter detectors that first remove ambient radon progeny, before filtering-out and counting new unattached radon progeny formed inside a large measurement volume under controlled conditions (Chambers et al., 2022; Griffiths et al., 2016; Whittlestone and Zahorowski, 1998), and (iii) direct electrostatic deposition monitors, which
also remove ambient progeny and allow new progeny to form inside a small measurement volume, but deposit these progeny electrostatically on a detector (Grossi et al., 2012; Pereira and Da Silva, 1989; Wada et al., 2013).

Continuous, long-term atmospheric radon measurements are currently performed worldwide using the three principles of measurement mentioned above. Maximising the value and utility of such large datasets across a range of applications requires a traceability chain for calibrations and standardised data processing techniques appropriate to each type of detector. Although,
there have already been some efforts to compare and harmonise radon measurements across the existing eclectic global net-work (Grossi et al., 2020; Xia et al., 2010; Schmithüsen et al., 2017) more attention needs to be given to preparation of a standardised protocol for retrieving the highest-quality, most directly comparable, atmospheric radon datasets from each kind of contributing instrument. Due to the distinct measurement principles of each instrument, tailored approaches are necessary to maximise consistency and comparability of datasets. For instance, indirect one-filter detectors require corrections for tube
loss, equilibrium (for inlets below 80–100 m a.g.l.), and exclusion of foggy/rainy conditions. While two-filter detectors are highly sensitive and independent of tube length, measurement height or weather conditions, the larger models (700 L and 1500 L) can have a 4–8 % uncertainty on individual field calibrations (unless a transfer standard is used) and require response time correction. Meanwhile, electrostatic deposition monitors need to dry their sample air (and/or correct for water vapor), remove or correct for thoron ($^{220}$Rn). Therefore, to maximise consistency and comparability across various instruments, it is essential
to establish a standardized processing procedure for each instrument type. This instrument-specific standardized procedure should be applicable to any atmospheric station measuring radon concentration with that type of instrument, and would en-able optimal utilisation of radon measurements by the atmospheric composition research community, particularly in studies verifying GHG emission estimates.

To this end, here we present a new protocol for processing measurements made by the fastest (30-minute temporal reso-
lution), most sensitive (detection limit ~0.025 Bq m$^{-3}$), and most widely used radon detectors within global and European atmospheric monitoring networks - the 1500 L "two-filter dual flow-loop" detector, developed by the Australian Nuclear Sci-ence and Technology Organisation (ANSTO). In the last 28 years, 50 two-filter radon detectors have been part of campaigns or global and European atmospheric monitoring networks (39 of which are still operational), 11 (4) of which are part of the ICOS (UK DECC) network. Compared to indirect detectors, two-filter detectors provide a measure of radon concentration that
is independent of height above ground, distance from land, meteorological conditions (e.g. fog/rain), fetch conditions, ambient





aerosol loading, length or type of sampling tube. Consequently, their calibration traceability is more readily achievable under a wide range of measurement conditions (Chambers et al., 2022).

This study utilizes one year (September 2020–August 2021) of radon measurements by 1500 L two-filter detectors at the three UK sites with contrasting sample inlet heights: TAC, HFD (UK DECC sites), and Weybourne Atmospheric Observatory (WAO, a UK ICOS and affiliated DECC site). The specific objectives are to: (i) outline the expected maintenance protocol for these detectors, (ii) outline a proposed standard processing protocol for near real time data use, including calibration, response time correction (Griffiths et al., 2016), standard temperature and pressure (STP) correction, and, optionally, correcting output to dry-air amount fractions; (iii) validate the timing of response time corrected radon concentrations using well-defined calibration events; and (iv) assess the accuracy of the resulting radon signal by comparing it with high resolution GHG ($CO_2$, $CH_4$, and $N_2O$) amount fractions measurements aggregated to 30-minute values.

## 2 Measurement sites and instrumentation

### 2.1 Measurement sites

This study focuses on atmospheric radon concentration and GHG ($CO_2$, $CH_4$, and $N_2O$) amount fraction measurements made at three sites of the UK DECC network with contrasting inlet heights (TAC, HFD, and WAO; Figure 1). Measurements at RGL (51.998° N, 2.540° W) were excluded due to the limited calibration and background measurement history within this study's time frame. Measurements at MHD were also excluded since radon concentrations at this station are conducted with a one-filter detector (Biraud et al., 2000). Each station in the UK DECC network measures at least $CO_2$, $CH_4$, $N_2O$ and sulfur hexafluoride ($SF_6$). However, some also measure carbon monoxide (CO), stable isotopic ratios ($\delta^{13}C(CH_4)$, $\delta^2H(CH_4)$), radiocarbon in atmospheric $CO_2$ ($\Delta^{14}CO_2$), and the oxygen/nitrogen ($O_2/N_2$) ratio.

For the results here, measurements of radon concentration, $CO_2$, $CH_4$, and $N_2O$ amount fractions have been aggregated to hourly temporal resolution and reported in local time (LT). The Northern Hemisphere seasonal convention has been adopted (i.e., autumn: September–November; winter: December–February; spring: March–May; summer: June–August).

### 2.1.1 Tacolneston (TAC)

The TAC tall tower (52.518° N, 1.139° E; 56 m above sea level (a.s.l.)) is situated southwest of Norwich, 28 km east of Thetford, in Norfolk. Sampling inlets are arranged at three heights: 54, 100 and 185 m a.g.l. Since 2012, measurements of $CO_2$, $CH_4$, $N_2O$, CO, and $SF_6$ amount fractions have been taken at all heights. In 2020, radon concentration started being sampled from 175 m a.g.l. Additional technical details regarding the tower setup can be accessed in Stanley et al. (2018). As well as being part of the UK DECC network, TAC is part of the Advanced Global Atmospheric Gases and Experiment (AGAGE) network and is a WMO GAW regional site.

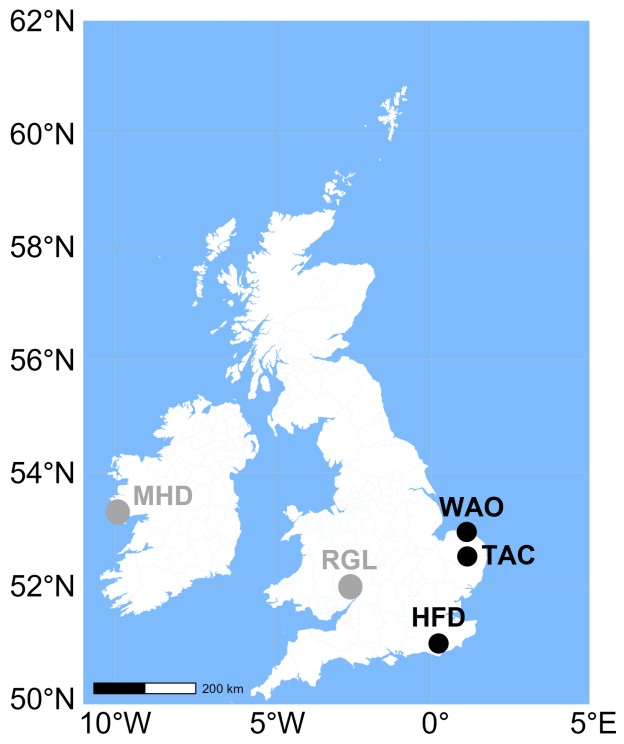

**Figure 1.** Geographical position of UK DECC stations measuring radon activity concentration: TAC, HFD, RGL, and MHD, along with affiliated WAO - an ICOS site. *Note: measurements from MHD and RGL, highlighted in grey, are not considered in this study.*

### 2.1.2   Heathfield (HFD)

The HFD tall tower (50.977° N, 0.231° E; 157 m a.s.l.) is situated in southeast England, 20 km from the coast, surrounded by woodland and agricultural green space (> 90 %). Measurements at this site are being conducted from an existing telecommunication tower. Sampling began in January 2014, and measurements of key GHGs ($CO_2$, $CH_4$, $N_2O$, CO, and $SF_6$) are being made from two sampling inlet heights (50 and 100 m a.g.l). Atmospheric radon measurements were introduced in 2020, with an independent inlet at 100 m a.g.l. Further information on technical details of the tower setup can be found in Stavert et al. (2019).

HFD is a part of the UK DECC network and is also a WMO GAW regional site. HFD is considered to be a background site of the UK DECC network as the predominant southwesterlies experience little land fetch prior to reaching HFD from the Atlantic Ocean (see Figure 1). However, it can also experience high pollution events since it is relatively close to the conurbation Royal Tunbridge Wells (17 km north-northeast), greater London (40 km north-northeast), and continental Europe (south-west).





### 2.1.3 Weybourne Atmospheric Observatory (WAO)

The WAO (52.951° N, 1.122° E; 17 m a.s.l.) is situated on the north Norfolk coast, approximately 35 km north-northwest of Norwich, 170 km northeast of London and 200 km east of Birmingham. WAO is an ICOS site, a WMO GAW site, and is also an affiliated UK DECC site. WAO was established in 1992 (Penkett et al., 1999) and a wide array of atmospheric gas
species (GHG amount fractions, stable isotopic ratio, reactive gases, as well as radon since March 2018) are measured there from a sampling inlet at 10 m a.g.l. Due to its location, WAO receives a variety of air masses from a range of sources including well-mixed background air (Atlantic, Arctic, North Sea) and polluted air (European, UK) (Adcock et al., 2023; Fleming et al., 2012; Forster et al., 2012).

## 2.2 Radon instrument: 1500 L dual-flow-loop two-filter

### 2.2.1 Operating principle

Atmospheric radon concentration is measured at all three sites (HFD, TAC, and WAO) using 1500 L dual-flow-loop two-filter radon detectors, designed and built by ANSTO, which provide half-hourly, high precision measurements. The principle of operation is described in detail elsewhere (Chambers et al., 2022; Griffiths et al., 2016; Thomas and Leclare, 1970; Whittlestone and Zahorowski, 1998) and is only summarised here. The detector relies on gross $\alpha$-counting, making the signal sensitive to
other radon isotopes (e.g., $^{220}$Rn, half-life of 55.6 s; and actinon: $^{219}$Rn, half-life of 4 s) as well. To eliminate contributions from unwanted radon isotopes (of which the longest lived is $^{220}$Rn), this system includes a "thoron delay volume" prior to the first filter, which acts to delay the sampled air for $\sim$ 5 minutes, allowing time for all short-lived radon isotopes to decay to their aerosol progeny. Following this, the sampled air is filtered (by the "primary filter') to remove the progeny of all radon isotopes and any ambient aerosols before it passes through to the main delay chamber, where a portion of the sampled ambient
$^{222}$Rn decays under controlled conditions to produce unattached aerosol progeny. These new unattached progeny (particularly the short-lived $\alpha$-emitters $^{218}$Po and $^{214}$Po) are then efficiently collected on a 20 $\mu$m stainless steel mesh (the second filter). The instrument then reports the number of $\alpha$-decays counted each 30 minutes using silver activated zinc-sulphide (ZnS(Ag)) scintillation material coupled to a photomultiplier (referred to as the "measurement head"). Output from the photomultiplier is amplified and fed into a discriminator, and the detector's net counts (lower level of discrimination - LLD) is determined from
the number of counts which lie within a pulse-height window (0.5–1.0 V) from which the activity concentration of radon (in Bq m$^{-3}$) is calculated. The timestamp associated with a count represents the end of the measurement period.

### 2.2.2 Instrument maintenance

Although ANSTO two-filter radon monitors are designed to require minimal maintenance, a degree of periodic maintenance is required to minimise ongoing operational costs and ensure consistent optimal performance, a pre-requisite for effectively
harmonising measurements across a network. Most crucial to ongoing performance is the detector's measurement head, which contains the second filter and a plastic sheet impregnated with ZnS(Ag) powder. These materials should be replenished every





5 years. The second filter will slowly accumulate lead ($^{210}$Pb, half-life of 22.3 years) and can not be cleaned. If the background becomes too high (e.g., $> 8$–10 counts min$^{-1}$), then the detection limit will deteriorate. The integrity of the ZnS(Ag) powder will also deteriorate over time (faster in very humid environments) and progressively reduce the detector's sensitivity to radon

(typically a change of 0.5–1.5 % per annum (p.a.)). If the detector is in a high vibration environment, there should be an annual check that the measurement head is still properly seated in the central pipe (following instructions in the detector's commissioning document). Likewise, if the detector is moved vigorously with the measurement head still inside, the seating of the head in the central pipe should be checked.

The next most important maintenance consideration is the detector's calibration system. If a pump is used to flush the radon

source (rather than compressed gas), the stability of the delivery flow rate (0.10–0.15 L m$^{-1}$) should be checked every 3–6 months or radon delivery during calibrations may become inconsistent. The desiccant tube attached to the calibration system inlet should be checked and refilled every 3 months to maintain consistent humidity levels. Fluctuations in humidity within the source capsule could potentially affect the radon emanation rate. Conversely, compressed gas calibration systems only require a 2-yearly replacement of the gas bottle for maintenance.

To protect the radon detector's sampling pump and extend the life of the primary filter, it is essential to prevent dust from passing too far through the sampling line. Typically, an easily accessible and cost-effective coarse aerosol filter is installed upstream of the pump. Depending on the expected aerosol loading at each site, this pre-filter should be inspected and/or replaced every 6–12 months.

The detector should be checked for leaks annually, with the leak rate not exceeding 2 L m$^{-1}$. To prevent near-surface

radon-rich air entering the detection volume due to potential small leaks, it's crucial to maintain the operating overpressure of the detector's main (1500 L) delay volume between 100–200 Pa. Note that if a long exhaust line is fitted to the detector the associated flow impedance may increase the overpressure to between 200–300 Pa. The inlet line (from the base of the sampling tower to the inlet of the detector) should also be checked for leaks annually.

The internal clock of the detector's data logger should be synchronised to the networked operating computer quarterly and

the logger's internal battery replaced every 5 years. All moving parts of the detector should be replaced every 10 years to avoid mechanical failure, and all electrical components (particularly power supplies) should be replaced every 10 years to prevent electrical noise developing.

### 2.2.3 Instrument calibration

ANSTO two-filter detectors are too large to be moved periodically and traceably calibrated in a controlled climate chamber, and

their high sampling rate (90 L m$^{-1}$) makes it financially and logistically impractical to attempt regular, traceable calibrations directly in situ (which would require a 90 L m$^{-1}$ flow of radon-free air for around 8 hours; either through a filtration system or a large bank of gas cylinders). The ideal calibration approach would be to transfer a traceable calibration to the operating detector using a mobile calibration transfer device (Chambers et al. 2022; Rottger et al 2023), but few suitable calibration transfer devices are currently available. The compromise usually adopted is to conduct approximate in situ calibrations using



a portable radon source while the instrument continues to sample ambient air. The procedure for a single calibration event is described below.

All ANSTO two-filter radon detectors in this study are calibrated using passive Pylon 2000A $^{226}$Ra sources ~49 kBq (Pylon Electronics, https: //pylonelectronics-radon.com/radioactive-sources/, last access: 24 March 2024; the source strength at each site is reported in Table A1). The timing of calibrations, duration of calibrations and source flushing times are all user-defined

and can slightly change from site to site.

Prior to initiating a calibration, the $^{226}$Ra source should be well flushed, with the exhaust directed to the outside ambient air at a point well removed from the detector sampling location. The flushing time required should be determined empirically for each installation and will depend on factors such as the source strength, the flushing flow rate, and the time since the source was last flushed. A period between 5–10 hours is usually sufficient. A calibration is then performed by continuing to flush

air through the source into the detector's sampling airstream for a period of 5 hours while the detector continues to operate normally (see Figure 2). After injection finishes, it can take up to 6 hours for the radon concentration inside the detector to return to ambient values.

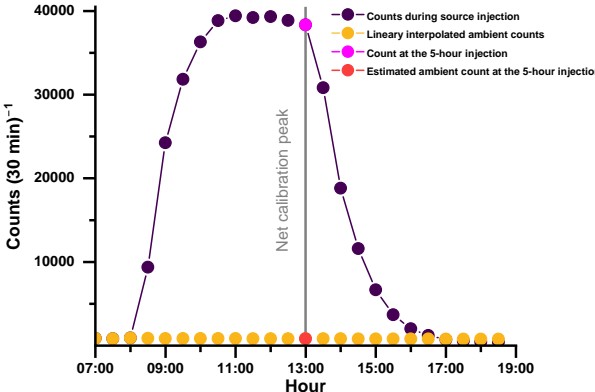

**Figure 2.** Example calibration peak resulting from a 5-hour injection and net calibration peak magnitude derived from linearly estimated ambient radon concentration at the time of peak counts.

If the source is being flushed by a pump, using dried ambient air, the flushing rate should not exceed 0.15 L m$^{-1}$ (to minimise the amount of ambient $^{222}$Rn and $^{220}$Rn introduced to the system). However, if the source is being flushed by radon-

free compressed gas, the flushing rate can be increased (e.g., 0.20 L m$^{-1}$) to improve the consistency of radon delivery. Once the calibration injection has been performed the calibration coefficient, $c_{cal}$ (counts s$^{-1}$ (Bq m$^{-3}$)$^{-1}$), is calculated through the following equation:

$$c_{cal} = \frac{(LLD_{peak} - LLD_{peak,a})}{1800} \times \frac{F_{ex}}{d_{source}} \times s \tag{1}$$





where $LLD_{peak}$ is the count recorded at the 5-hour injection, and $LLD_{peak,a}$ is the estimated ambient count (at the 5-hour

peak counts), $F_{ex}$ (m$^3$ s$^{-1}$) is the sampling flow rate, $d_{source}$ (Bq s$^{-1}$) is the radon delivery rate of the $^{226}$Ra calibration source, and $s$ is a dimensionless scaling factor.

The scaling factor is required to account for the fact that after only 5 hours of injection from the source, the radon concentration within the detector will not have completely come to equilibrium, due to its logarithmic growth curve. Based on a model of the detector response (Griffiths et al., 2016) after 5 hours of injection into a 1500 L radon detector sampling at 90 L m$^{-1}$ the

concentration of radon inside the detector would have reached ∼99.3 % of the equilibrium value. This leads to scaling factor of 1.007 (reciprocal of ∼99.3) for a 1500 L detector. This scaling factor will vary with the length of calibration injection and sampling flow rate.

Assuming that the calibration unit is functioning reliably, the largest source of uncertainty in the field calibration process is in the estimation of $LLD_{peak,a}$. Typically, an assumption is made that ambient radon concentrations change little, and linearly,

over the duration of the calibration event (10–11 hours;yellow line, Figure 2). However, the accuracy of this assumption is influenced by many factors, including timing of the calibration injection within a diurnal cycle, ambient wind speed during the calibration event, air mass fetch conditions, and day length (see Figure 7a in Chambers et al. (2022)). Consequently, at flat inland sites calibrations are best performed during windy conditions and the source injection should be timed to finish around 13:00–14:00 LT, when the boundary layer is deepest and most well-mixed. For coastal or island sites calibrations are best

performed under windy conditions with oceanic fetch. Under oceanic fetch conditions the calibration injection duration can be increased to 6 hours to reduce the magnitude of the required scaling factor, $s$. At high mountain sites calibrations are best conducted at night under katabatic flow conditions. Either calibrations can be initiated remotely during suitable conditions, or they can be set on a regular schedule, and events occurring during non-ideal conditions can be excluded.

To minimise relative uncertainty introduced by the $LLD_{peak,a}$ estimation process, the detectors are usually calibrated at a

radon concentration that is at least an order of magnitude greater than the expected annual maximum ambient radon concentration at the site. Despite being calibrated at relatively high radon concentrations, the resulting calibrations have been proven to be quite linear down to very low radon concentrations (Röttger et al., 2023).

To this end, consistent, application specific calibration approaches need to be agreed upon and formalised (a key goal of this study) rather than users at different sites simply applying each monthly calibration coefficient. As previously mentioned,

if a calibration transfer device is available (Chambers et al., 2022; Röttger et al., 2023) then a traceable calibration can be transferred to an operating 1500 L detector in situ, over a period of around 2 weeks, without the need for approximate monthly calibrations. In this case, it would only be necessary to calibrate the detector once per year to characterise the slow change in sensitivity over the 5-year period until the measurement head is refreshed. If one portable calibration transfer device was allocated to a network of five 1500 L two-filter radon detectors, the cost would be less than buying individual calibration

devices for the 5 large detectors.

In the absence of a calibration transfer device, the next most accurate calibration approach (as described in Chambers et al. 2022) is to develop a linear calibration model for the 5-year period between detector head replacements, since the gradual degradation of the ZnS(Ag) scintillation material is the primary cause of changing detector sensitivity. However, this calibration



method can only be applied retrospectively. While the level of calibration accuracy provided by this approach is necessary for
deriving consistent vertical radon gradient measurements from tall towers, an alternative calibration approach (with a slightly
increased uncertainty) is necessary if the observed radon concentrations need to be used in near real time (described in Section
3.2 and 3.3).

### 2.2.4  Instrumental background

The instrumental background, $LLD_{bg}$ (counts per 30 minutes), of ANSTO two-filter detectors arises from various contributing
factors, such as: (i) cosmic radiation, (ii) natural radioactivity of surrounding rocks, soils or building materials, (iii) accumula-
tion of $^{210}$Pb as well as intrinsic background count rate of the photomultiplier caused by other effects (e.g., photons (Wright,
2017)) on the detector's second filter, and (iv) self-generation of radon (by trace amounts of $^{226}$Ra in the detector building
materials). The first two factors are small, site specific and relatively constant. $^{210}$Pb accumulation gives rise to an increasing
$\alpha$-count (due to subsequent $^{210}$Po decay). Self-generation of radon inside a detector is also constant (due to the 1600-year half-
life of $^{226}$Ra), typically small, and varies from build-to-build of each detector. Considered over multiple years, the detector's
background count increases approximately linearly.

Ideally, the background should be determined while the detector is operating normally, sampling radon-free air at 90 L m$^{-1}$
for a period of at least 10–12 hours. However, the necessary supply of radon-free air for regular tests of this kind is logistically
and financially impractical. The compromise for field $LLD_{bg}$ is to simply shut the detectors down for a 24-hour period. It is
advised to conduct these checks every two to three months. Background measurements are conducted by deactivating blowers
in both the external and internal flow loops, as well as closing the detector's inlet solenoid valve for 24 hours. This process is
divided into three stages:

1. **Decay (5.5 hours)**: This 5.5 hours period marks the decay of the short-lived radon progeny on the detector's second filter
   below detection limits within 5–6 hours of the detector shutdown.

2. **Background (18.5 hours)**: The count reading stabilises (with a degree of uncertainty) to a constant background level.

3. **Recovery (1–2 hours)**: Upon reactivating the blowers and detector's inlet solenoid valve is opened, the instrument
   undergoes few measurement cycles (1–2 hours) to readjust itself and return to the ambient levels

During a background check the detector's inlet is blocked to prevent flow through the detector arising from venturi ef-
fects across the inlet line (near the top of the sampling tower). However, the detector's exhaust valve is left open, making it
possible for back diffusion of radon to occur into the detector. This is usually only a concern at sites where nocturnal radon
concentrations at the detector's location are high. Closing both the inlet and exhaust valves of the detector during a background
measurement is not advisable unless the detector is in a temperature-controlled environment or equipped with a pressure relief
mechanism. Diurnal temperature variations could lead to pressure fluctuations within the detector, potentially causing leaks.
The inlet valve is closed in preference to the exhaust valve during a background check to prevent over-pressuring the detector
if the powerful external flow loop blower is accidentally restarted before the exhaust valve is opened.




If the 18-hours background data in step 3 above is observed to gradually decrease, rather than being approximately constant, $^{220}$Rn contamination of the second filter (in the measurement head) is likely. This is indicative of a leak (e.g., in the sampling line or the detector), or an accumulation of $^{226}$Ra containing dust in the detector's thoron delay volume.

## 2.3 Greenhouse Gas (GHG) instruments

At HFD, continuous, high frequency (0.2 Hz) $CO_2$ and $CH_4$ measurements are made using a G2401 cavity ring-down spectrometer (CRDS; Picarro Inc., USA), while (0.4 Hz) $N_2O$ measurements are made using G5310 CRDS. At TAC, (0.3 Hz) $CO_2$ and $CH_4$ measurements are made using a G2301 CRDS, while (1 Hz) $N_2O$ measurements are made using an off-axis integrated cavity output spectrometer (OA-ICOS; Los Gatos Research Inc., USA). Measurements are made at different inlet heights: HFD at 50 and 100 m a.g.l and TAC at 54, 100 and 185 m a.g.l. (see Stanley et al. (2018)). Sampling occurs alternately

at different inlet heights along the tower, resulting in measurements from various heights not being simultaneous. For instance, air is sampled for about 20 min at 50 m a.g.l., followed by another 40 min of sampling at 100 m a.g.l. with a minute of flushing in between to avoid contributions from the previous level. To address this height-switching during sampling, measurements are interpolated to create a consistent hourly time series instead of straightforwardly averaging over an hour. Data are corrected for daily linear instrumental drift using standard gases (natural composition) and for instrumental non-linearity using cali-

bration gases (natural composition). Further information on instrumentation, including flow diagrams, measurement protocol, calibration of standards and uncertainty estimation can be found in Stanley et al. (2018) and Stavert et al. (2019).

At WAO, 1-minute measurements of $CH_4$ and $N_2O$ are made using a commercial Fourier Transform Infra-Red Spectrometer (FTIR) instrument (ACOEM Spectronus$^{TM}$): a detailed description of the FTIR can be found in Hammer et al. (2013). The instrument is routinely calibrated with gases provided by the ICOS Flask and Calibration Laboratory (FCL) and amount

fractions are traceable to the WMO calibration scales (Crotwell et al., 2017; Yver-Kwok et al., 2021). While $CO_2$ amount fraction is measured every second and averaged to 2-min by a non-dispersive infrared (NDIR) analyser from Siemens Corp. Full details on instrumentation, measurement protocol and calibration strategy can be found in Adcock et al. (2023).

## 3 Processing of atmospheric radon data: Towards a standard protocol

### 3.1 Detector control and data collection

Operation of the ANSTO detectors, their data logger and calibration units are controlled by Radon Detector Monitor (RDM) software installed on local site computers. For PC operating systems prior to Windows 10, a Visual Basic GUI version of RDM was distributed. For Windows 10 or later, as well as for Linux systems, a Python-based GUI version of RDM has been available since August 2022 (https://github.com/anstoradonlab/radon-monitor/releases/, last access: 24 March 2024). RDM is responsible for the collection, display and storage of all raw detector output (see Figure 3 and elaborated further in Table

A2) and based on user-defined parameters, maintains full control of scheduled calibration and background events. However, calibrations and backgrounds can also be remotely reconfigured and initiated if the computer is networked.





All measurement and diagnostic quantities associated with the radon detector operation are polled by the internal Campbell Scientific CR800 data logger every 10 seconds, then totalled or averaged every 30 minutes. Totalized counts include LLD, upper limit of discrimination (ULD – electrical noise), and sample flow rate, while all other parameters are averaged (see

Figure 3).

RDM retrieves data from the logger every 30 minutes and saves monthly files in two formats: CSV and SQL database files. The CSV files only contain 30-min records of raw measurement and diagnostic quantities, kept small (<140 kB month$^{-1}$) for ease of remote file transfer. The database files contain all 10-second and 30-minute raw and diagnostic quantities, full status and operational information from the calibration system, and a complete log of system and error messages (<2 MB month$^{-1}$).

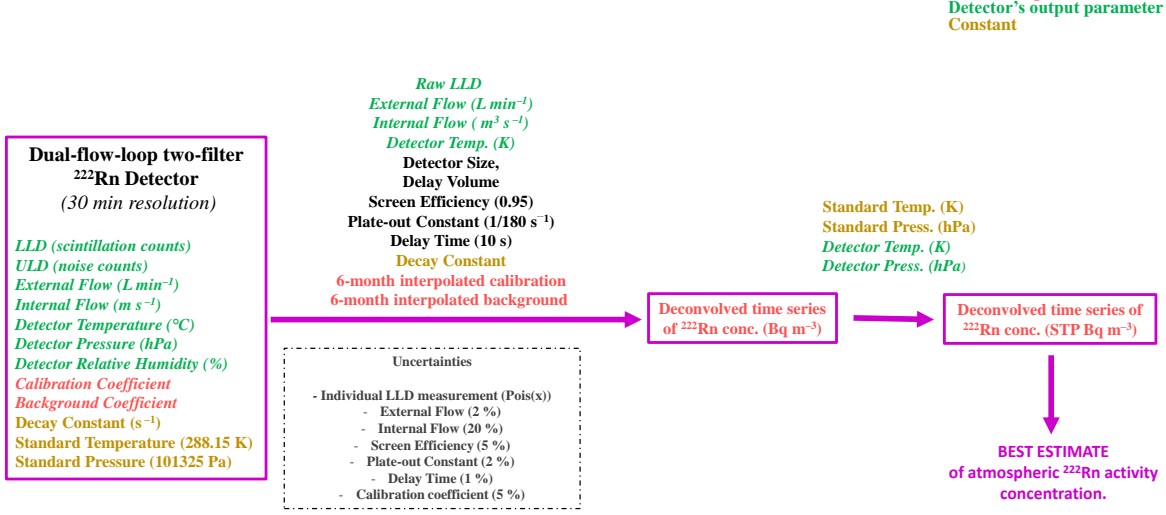

**Figure 3.** Workflow to calculate the best estimate of radon activity concentration: This flow illustrates the processing steps from measured "raw" detector output through calculated parameters and constants to derive the best estimate of radon activity concentration, described in detail in section 3. The parameters which influence the uncertainty derived from the deconvolution process, are also highlighted (described in subsection 3.7).

## 3.2   Data quality control, background determination and calibration

The first step of quality control is to check and correct any data timestamping errors. Issues such as power supply disruptions, logger malfunctions, or communication errors can lead to missed, duplicated, or incomplete data records.

**LLD counts:** The internal and external flow rate, ULD, high voltage, and tank pressure are the critical factors that give the indication whether the LLD count is going to be flagged as valid or invalid. For a given time point, if these parameters varied





beyond designated site-specific limits determined to reduce their accuracy, LLD is rejected. LLD was rejected too if it was lower than instrumental background.

The ratio of detector volume (e.g., 1500 L) to sampling flow rate (external flow loop; L min$^{-1}$) determines the time that sampled air is delayed inside the detector (during which some of the sampled $^{222}$Rn will decay). The delay time should be between 15–20 minutes, necessitating a flow rate between 75–100 L min$^{-1}$.

The flow rate of the internal flow loop should be sufficient to exchange all the air within the detector through the measurement head in less than 3 minutes (the half-life of $^{218}$Po). While a flow rate of around 5.5 m s$^{-1}$ is technically sufficient for this purpose, a faster rate is desirable, and values of 6–12 m s$^{-1}$ are typically achievable (based on individual blower performance and flow impedance of the measurement head).

The main detector volume is kept at a slight positive pressure with respect to ambient (described in more detail in 2.2.2) to
prevent near-surface air entering the detector should any leaks develop. A large leak would reduce this pressure and a system blockage downstream would increase it. The micro mass flow controller used to estimate this overpressure has a mV output (prone to some calibration drift). An option exists to enter user defined calibration coefficients for this sensor (ideally updated annually) to retrieve output in Pa, but raw output in the range 2200–2400 mV corresponds approximately to an overpressure of 100–120 Pa.

The sensitivity of the photomultiplier tube in the measurement head changes with the supply voltage. The operating high voltage is unique to each detector counting system and is determined when the detector is commissioned (and rechecked if any of the detector electronics are changed). Once set, this value should not be allowed to change by more than $\pm 10$ V without being manually readjusted back to the nominated value.

The counting system of two-filter detectors is sensitive to electrical, electro-magnetic or some radio-frequency noise. Ideally
the detectors should be operated on their own power circuit or through a UPS, and large electrical motors (e.g., pumps, compressors) should not be operated nearby. These forms of noise typically result in raw counts of a higher voltage than counts due to $\alpha$-decay. A voltage discrimination threshold of 1.0 V is set within the counting electronics to distinguish between raw $\alpha$-counts from radon progeny (LLD), and noise counts (ULD). Ideally, the ULD counts represent the number of LLD counts that are due to noise. When few noise counts are present (e.g., $\leq 1$ counts min$^{-1}$; site specific), LLD counts can still be repre-
sentative raw radon count. Notably, at TAC, ULD counts consistently ranged between 3–7 counts min$^{-1}$ throughout the entire measurement period. It is important to note that this noise originated from the pump of another instrument. Discarding all measurements made it impractical to implement flagging in this context. Nevertheless, ongoing efforts are being made to develop solutions aimed to mitigate the noise, alongside the ongoing evaluation of uncertainties associated with these noise counts.

As an example, Table 1 summarises the designated acceptable limits of diagnostic quantities for the HFD 1500 L radon
monitor.

The final consideration for flagging the LLD data are background, calibration, and maintenance events, or power failures. The detector is out of operation for the entire 24-hour background check, as well as up to 2 hours after the detector restarts. The detector is also out of operation for the 5-hour source injection period of a calibration, as well as the 6-hour period required to flush the enhanced radon out of the detector. During maintenance periods there are higher risks of $^{220}$Rn contamination





**Table 1.** Quality check parameters for LLD flagging: Critical factors (internal and external flow rate, ULD, high voltage, differential pressure) with minimum and maximum values – Highlighted through HFD site specific thresholds.

| Parameter | Min value | Max value |
|---|---|---|
| External flow (L m$^{-1}$) | 75 | 100 |
| Internal flow (m s$^{-1}$) | 7 | 10 |
| Voltage (V) | 720 | - |
| Differential pressure (mV) | 2000 | 2600 |
| Upper limit of discrimination (count (30 min)$^{-1}$) | - | 35 |

(if the detector has been opened) or diagnostic parameters being out of range. After power failures it can take the detector 1–2 hours to return to normal monitoring conditions. All such periods should be flagged out of the final raw dataset prior to further processing.

**Instrumental background event:** Once background checks had been initiated as outlined in section 2.2.4 and other diagnostic parameters (high volatge and noise; Table 1) were verified to be within acceptable ranges, each background (Figure 4a–c)

was processed as follows: the initial 6 hours of the 24-hour period were removed; the last 30-minute sample was excluded if the blowers restarted early; and a check was made that the remaining data were linear (approximately constant) with relatively low variance. The median of these values was then taken to be the background reading.

During the measurement period presented here, an important observation is that each of the three sites depicted in Figure 4a–c experienced an instance where a background event was not recorded. In each case this missing background event was labeled

"assigned background", for which the method of determination is explained in the next section 3.3. Typically, instrumental background checks are automated and scheduled quarterly. However, a deviation from this established routine sometimes occurred due to a software crash, leading to a significant gap in the background check process.





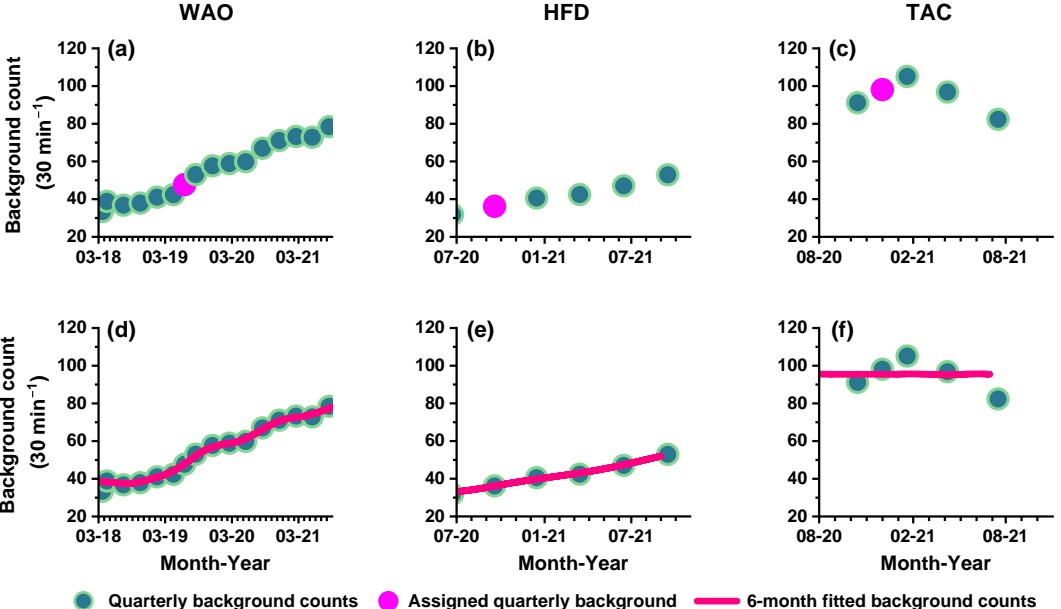

**Figure 4.** Instrumental background events for three sites: a) WAO, b) HFD, and c) TAC, covering the period from the initiation of measurements until August 2021. Assigned background events are highlighted (see section 3.3 for details of assigned values). Corresponding 6-month fitted instrumental background are shown in d) WAO, e) HFD, and f) TAC (see section 3.3 for details of fitting).

**Calibration event:** The calibration plan for all radon detectors in this study was for monthly calibrations (scheduled in RDM to occur on the same day of each month) such that the calibration peak ($LLD_{peak}$) occurred between 13:00–14:00 LT,

to coincide with a deep, well-mixed boundary layer. Consequently, any calibration events for which $LLD_{peak}$ did not occur within this time window were rejected. All sites in this study also had regular exposure to "baseline" atmospheric conditions (medium-to-long term oceanic fetch conditions, with low variability in radon concentrations). Consequently, a site-specific threshold value was set for the ambient radon count representing the upper limit of baseline variability (50 counts min$^{-1}$), and any calibration event for which the estimated $LLD_{peak,a}$ exceeded this value was flagged.

Derived monthly calibration factors for all three sites (WAO, HFD, TAC) are presented in Figure 5a–c, covering the period from the initiation of measurements until August 2021. At HFD (Figure 5b), the first 6 months were flagged due to sampling flow problems caused by a partial blockage of the inlet line. Other causes of flagged calibration events in Figure 5a–c, were attributed to the threshold exceedance of $LLD_{peak,a}$, calibrations occurring at night due to software crashes, and insufficient flushing of the source. For example, the last three calibration events at TAC (Figure 5c, June, July and August 2021) were all

too high due to a poorly flushed source. The source flushing time at all sites was originally set to 5 hours (typical for 20 kBq $^{226}$Ra sources). However, all sources in this study had activities >30 kBq $^{226}$Ra (see Table A1) and it was determined that effective flushing of the source capsules required flushing periods lasting 8–12 hours at 0.15 L min$^{-1}$. Older-style calibration





systems were used in this study (using pumps and needle-valve flow control, not compressed gas and a mass flow controller), so it was important to regularly check and correct the source's flushing.

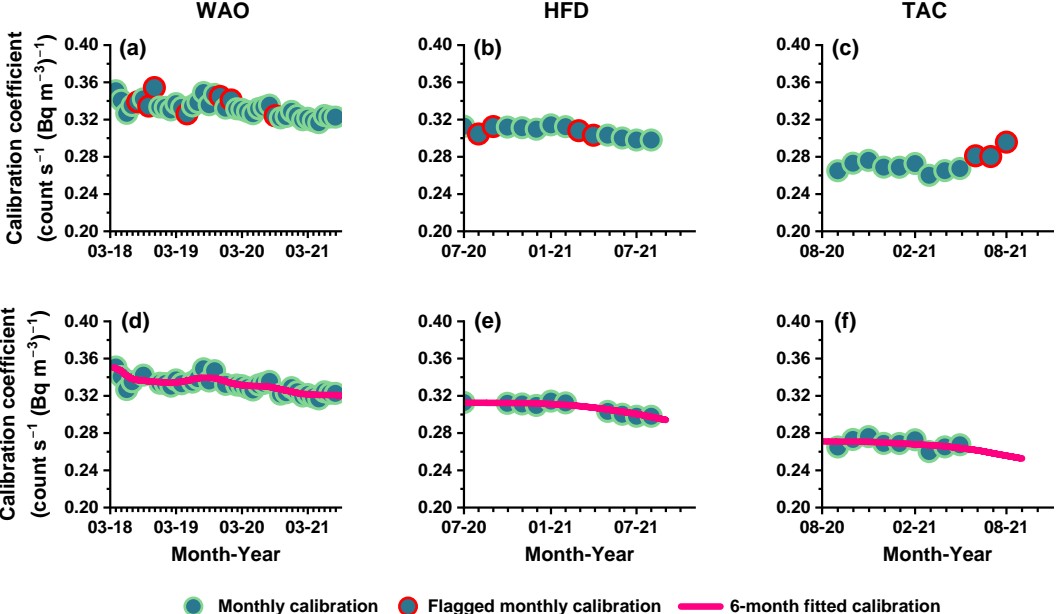

**Figure 5.** Monthly calibration coefficients for three sites: a) WAO, b) HFD, and c) TAC, covering the period from the initiation of measurements until August 2021. Flagged calibration events are highlighted (see section 3.2 for exclusion criteria). Corresponding 6-month fitted calibration coefficients are shown in d) WAO, e) HFD, and f) TAC (see section 3.3 for details of fitting).

## 3.3 Data continuity: Fitting instrument calibration and background

To ensure a continuous time series and to avoid unnecessary step changes, the Savitzky-Golay fitting method is used on successive subsets of calibration coefficients and instrumental background counts. This method involves the following steps:

1. **Fitting window size:** Calibration and background values are selected within a specific range. This range is defined by two half-regression windows, each with a duration of 92 days.

2. **Fitting:** Within the defined window, a curve is fitted through the selected values using a $1^{st}$ order polynomial - linear fit, based on the data points available in the defined window.

3. **Smoothing window size:** The result of the moving fit requires further smoothing. Centre moving average is used with a defined two half-smoothing windows, each with a duration of 92 days.

4. **Smoothing:** The centre moving average is performed.





5. **Handling edge data:** To ensure a smooth transition at the beginning and end of the dataset, the data before the first LLD count and data after the last LLD count are supplemented with the corresponding fitted values from these points.

In the rolling fit process, it is crucial to compare the current fitting with the preceding one, particularly in periods of overlap. In this regard, two types of data ranges are recommended: (i) the data range for selecting raw data and performing the fit, and (ii) the data range where the fitted data will actually be utilized. Additionally, the effectiveness of the Savitzky-Golay fitting method can be affected by gaps within the defined window. This issue is evident across three sites, with distinct gaps in background counts observed (see Figure 4). To address this, the concept of "assigned" values is introduced. First, data gaps larger than either the defined half-window size for fitting or a predetermined maximum gap are identified. The number of points to be inserted into these gaps is then determined using the ceiling function:

$$n = \left\lceil \frac{t_1 - t_0}{\text{gap}_{\text{max}}} \right\rceil \tag{2}$$

where $t_1$ and $t_0$ are the time coordinates at either end of the gap. The resulting number of data points to be input are inserted evenly within the gap, with spacing adjusted to the nearest 30 minutes. Finally, linear interpolation, based on the values at the gap's ends, is performed to assign values to these imputed data points.

The bottom panel of Figures 4d–f and 5d–f display fitted instrumental background ($f_{\text{bg}}$) and the fitted calibration coefficient ($f_{\text{cal}}$) for three sites: WAO, HFD, and TAC. The fitted coefficients reveal a site-dependent annual reduction in detector sensitivity, varying from 3.2 % to 4.7 %. At WAO, situated along the coast with the lowest source strength (see Table A1) and a sampling inlet, the calibration coefficient experienced a 3.2 % p.a. decrease. Similarly, TAC, with the highest sampling inlet, also witnessed sensitivity reductions of 3.3 % p.a., although it recorded the highest background rates among the three sites. The main reason of increased background is the ULD noise but although other factors may also contribute such as $^{220}$Rn contamination at this site and ingrowth of radon from $^{226}$Ra inside the detector . In contrast, the HFD detector exhibited a notably faster sensitivity decrease of 4.7 %, accompanied by the lowest background counts. The acceleration of reduction in sensitivity and increase in background rate will vary for each instrument, as every batch of materials used for making components has distinct levels of trace contamination of $^{226}$Ra and different consumable materials within the detector head, specifically the ZnS(Ag) scintillation material.

### 3.4 Instrument response time correction

The operational principle of two filter detector, briefly discussed in section 2.2.1, inherently causes a delay on reporting the true LLD signal. Approximately 40 % of the signal is observed to arrive one hour after the radon pulse is delivered (Griffiths et al., 2016). This delay necessitates time-response correction, particularly when employing sub-diurnal radon measurements for quantitative analysis or comparison with fast-response instruments like GHG ones. Griffiths et al. (2016) explored methods to correct for the detector's slow response, highlighting the effectiveness of a Bayesian approach using a Markov chain Monte Carlo sampler — a methodology employed here for deconvoluting the radon time series. The aim of deconvolution is to estimate the true signal within the temporal context while preserving the radon concentration levels. The deconvolution is





performed on the LLD counts, where all flagged LLD are removed prior to this step. Along with LLD counts, fitted calibration coefficient and instrumental background, the deconvolution routine also requires several output parameters: both external and internal flow rates, the detector's temperature and physical detector's parameters: its size, and the thoron delay volume. The

fitted instrumental background values are subtracted from the deconvolved LLD counts ($LLD_{dec}$). Following this subtraction, the $LLD_{dec}$ are then processed to calculate the activity concentration (Bq m$^{-3}$) of $^{222}$Rn, the equation is as follows:

$$^{222}\mathrm{Rn} = \frac{(LLD_{dec} - f_{bg})}{1800 \times f_{cal}} \tag{3}$$

For a given timestamp in the time series, the median of the deconvolution result at that timestamp is reported as the radon concentration of that particular timestamp.

**3.5 Standard reference temperature and pressure correction**

The last step of data processing involves normalising the deconvolved concentration of radon to standard reference temperature ($T_{stp}$) and pressure ($p_{stp}$) conditions (Bq m$^{-3}$ STP). For our case, we adapted the International Standard Atmosphere (288.15 K and 101325 Pa). This standard was selected mainly due to its representative average, as 15 °C approximates the average annual ambient temperature of the Earth's surface in temperate regions and makes it easier to be compared across different

regions and times of the year. This normalisation is achieved through a correction term as follow:

$$^{222}\mathrm{Rn} = \frac{(LLD_{dec} - f_{bg})}{1800 \times f_{cal}} \times \left( \frac{T_{inst}}{p_{inst}} \times \frac{p_{stp}}{T_{stp}} \right) \tag{4}$$

where ($T_{inst}$) and ($p_{inst}$) are temperature and pressure inside the detector's main delay volume.

This correction removes sensitivity to measurement height, facilitating comparison with modelled activity concentrations, and ensures that observed trends in radon concentration are solely due to environmental variations in mixing ratio brought

about by changing air mass fetch and transport times.

Regarding the harmonisation of radon observations across European and global networks, it should be noted that some atmospheric radon monitors operating within ICOS and other networks (e.g., the HRM, (Levin et al., 2002)), automatically implement an STP correction that assumes different pressure and temperature reference values ($T_{stp}$=273.15 K, $p_{stp}$=100000 Pa). Other monitors (e.g., the ARMON, (Grossi et al., 2012)) do not measure pressure and temperature directly within the instru-

ment delay volume and can therefore only make approximate STP corrections.

**3.6 Atmospheric water vapour corrections**

Water is a volatile component of the atmosphere. As such, water vapor can vary rapidly in space and time. Through the dilution effect, these changes can influence observed concentrations of other atmospheric constituents. Some radon monitors in European networks (e.g., the ARMON) dry sampled air prior to analysis. Furthermore, modelled radon values are usually

reported as dry air mole fractions. For harmonisation and intercomparison purposes, radon monitors should provide the ability to correct for water vapour effects.





ANSTO radon detectors measure the pressure, temperature, and relative humidity (*RH*) of air in the main delay volume, directly adjacent to the detector measurement head, providing a pathway for water vapour correction if required. However, the process of accurately converting *RH* into water vapor pressure involves navigating several challenges, including temperature

dependencies, measurement accuracy, environmental variability and assumptions in calculation equations. One commonly used method for this conversion is the Clausius-Clapeyron equations, refined with the Magnus approximation, with constants from Alduchov and Eskridge (1996). The equations are based on certain assumptions and empirical data, which might not reflect all environmental conditions, therefore introducing systematic errors. Another critical aspect to consider is that calibration of the radon detector is not based on dry air (the ambient air is used to flush the source), complicating the direct comparisons.

To assess the necessity for correcting water vapor, a simple calculation was carried out to compare the difference between dry and wet air across a spectrum of UK extreme climate conditions, as indicated in the Table 2. The maximum discrepancy observed was in scenario with high temperature and pressure of 7.7 %. The discrepancy was minimal (0–2 %) for air temperature spanning 0 to 30 °C. Considering the minimal impact of correcting the water vapor and potential for such correction to introduce noise or errors through another layer of data analysis, especially after the deconvolution process, this correction was

omitted. However, detector temperature and RH are reported along with radon concentration so that data users can make this correction themselves, if necessary.

**Table 2.** Comparison of dry and wet air across UK extreme climate conditions: evaluating the need for water vapor correction.

| Scenario | $p_{air}$ (hPa) | $T_{air}$ (°C) | RH (%) | $e_s$ (hPa) | $f_{wet}$ | $f_{dry}$ |
|----------|-----------------|-----------------|--------|-------------|-----------|-----------|
| 1 | 1053.6 | 40.3 | 100 | 74.93842 | 0.9558 | 0.8879 |
| 2 | 1053.6 | -27.5 | 100 | 0.644756 | 1.2197 | 1.2189 |
| 3 | 925.6 | 40.3 | 100 | 74.93842 | 0.8397 | 0.7717 |
| 4 | 925.6 | -27.5 | 100 | 0.644756 | 1.0715 | 1.0707 |
| 5 | 1013.2 | 15.0 | 100 | 17.01983 | 1.0000 | 0.9832 |

**Note on formulas used:** The wet air correction factor ($f_{wet}$) is calculated with the formula $f_{wet} = \frac{T_{stp}}{p_{stp}} \times \frac{p_{air}}{T_{air}+273.15}$; Similarly, the dry air correction factor ($f_{dry}$) employs the formula $f_{dry} = \frac{T_{stp}}{p_{stp}} \times \frac{p_{air}-e_{s,RH=100}}{T_{air}+273.15}$.

### 3.7 Combined measurement uncertainty

The uncertainty associated with the reported radon concentration is derived from the posterior distribution resulting from the

deconvolution process. This uncertainty is influenced by:

- **Gross alpha counting:** Governed by a Possion distribution, this uncertainty reflects the statistical variability inherent in counting process.

- **Flow rate variability:** The external flow rate introduces an uncertainty of 2 %. Variations in the external flow rate, especially rapid changes, can significantly affect the response time of the detector. Griffiths et al. (2016) demonstrated

that the deconvultion tends to over-correct the time series if the external flow rate is halved. The internal flow rate contributes a substantial uncertainty of 20 %. Optimizing the internal flow rate is critical for a few reasons: (i) to account





for the rapid decay of radon's first $\alpha$-emitting progeny ($^{218}$Po, half-life of 3.1 min), and (ii) to minimize the time for plate-out of unattached radon progeny on the internal surfaces of the detector. To further reduce opportunities for the plate-out of radon progeny inside the detector, a flow homogenization screen is implemented to reduce strong turbulent mixing inside the detector's main delay volume. This screen ensures a more laminar (plug-like) flow towards the measurement head.

- **Plate-out effect:** An absolute uncertainty of 2 % is given to plate-out effect.

- **Screening efficiency:** A 5 % absolute uncertainty is associated with the screening efficiency on homogenizing airflow and reduce turbulences.

- **Delay time:** The delay time represents the lag in system, with an absolute uncertainty of 1 %, and can arise from various sources such as: extended inlet lines, synchronization of the datalogger etc. While this delay time does not significantly impact the deconvolution routine, it will becomes critical when aligning the detector's response time with large, brief spike of counts.

- **Field calibration coefficient:** The calibration coefficient, with an overall uncertainty of 5 %, incorporates several aspects: the $^{226}$Ra source accuracy with an absolute uncertainty of 4 % and additional 1 % uncertainty arises from variation in radon production rate, scaling factor and the decay constant.

In addition to the above-mentioned uncertainties, there are other significant uncertainties that, although not currently addressed, should be considered in future analyses. These include uncertainties related to STP corrections, which typically stem from the accuracy, calibration, and response time of temperature and pressure sensors. A critical area to address is the uncertainty associated with field calibration of 1500 L detectors, that involves interpolating ambient counts (see 2.2.3). Although this problem can be eliminated by employing a portable calibration transfer standard detector (Chambers et al., 2022; Röttger et al., 2023). Another improvement in the uncertainty quantification includes instrumental background and calibration events themselves, alongside the "assigned" values (see 3.3), with the latter one having higher uncertainties.

For reporting, half of the difference between the 16th and 8th percentile of the deconvolution result for a particular timestamp is taken as the uncertainty of the radon concentration.

## 4   Assessing accuracy of the best radon activity concentration estimate

Two approaches are used to evaluate the accuracy of the deconvolved radon concentration. The first approach, under "controlled conditions" employs calibration events to validate both the absolute achieved concentration and the temporal alignment between estimated and measured radon values. The second approach, based on "real-world conditions", directly compares estimated radon concentrations with the GHG amount fractions obtained from fast-response detectors.





## 4.1 Validating radon activity concentration estimates under controlled conditions

Figure 6a–b depicts calibration events on July 2021 from two locations: HFD and WAO. In each case, radon concentrations were measured following a direct 5-hour injection from a $^{226}$Ra source ($^{222}$Rn$_{ini}$), and then the deconvolution algorithm was applied ($^{222}$Rn).

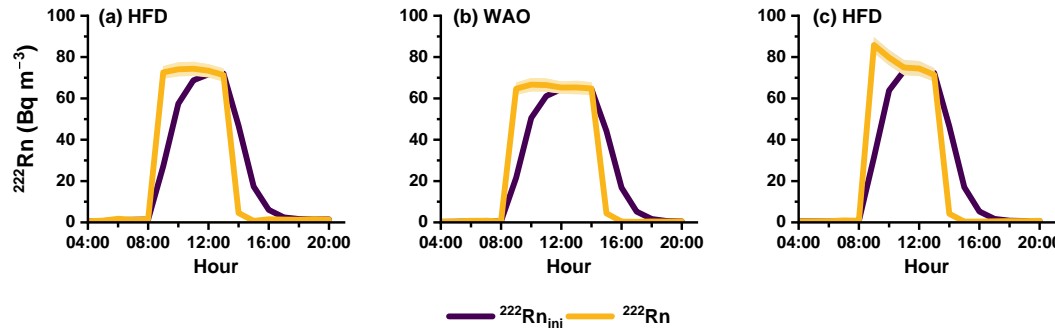

**Figure 6.** Comparison of initial and best radon concentration estimate (with the 16th–84th percentile range shaded) during a calibration event with a constant delivery radon concentration in July 2021 at: a) HFD and b) WAO, alongside a calibration event with non-constant radon delivery in August 2021 at c) HFD.

Specifically in the case of these events, when the deconvolution algorithm was applied some adjustments were necessary to enhance the algorithm's efficiency. Firstly, because the calibration stream of radon was injected directly into the detector (not at the sampling inlet point), the thoron delay volume value was set to zero. Secondly, since environmental atmospheric radon concentrations do not usually exhibit abrupt changes of the magnitude seen during calibration events, the smoothness constraints of the deconvolution routine were adjusted to accurately track the sudden change in radon concentration as the calibration source is turned on and off.

The purpose of the deconvolution algorithm is to correct the delay between the sampled radon concentration and the detected signal. In the case of a defined calibration injection period from a well-flushed source, the target result is essentially a square wave (bearing in mind that the detector reports concentrations at the end of each measurement period and continues to sample ambient air throughout the injection period). As shown in Figure 6a–b, despite the source injection being initiated at 08:00 LT at both sites, the signal from both detectors (yellow line) had only reached around 85 % of its target value by 09:00 LT. Similarly, after injection stops (13:00 LT at HFD, 14:00 LT at WAO), the signal from each detector takes around 3 hours to return to near ambient. By comparison, the deconvolved signal (purple line) exhibits a full increase in concentration within the first hour (the temporal resolution of these measurements) and shows an almost complete return to ambient concentrations in the hour after injection stops. The observed behaviour of the response time corrected data demonstrates the effectiveness of the deconvolution process in recreating sudden, unrealistically large, changes in radon concentration, lending credence to the claimed precision and reliability under real-world scenarios.





As evident in Figure 6a–b, when the deconvolved radon signal from a calibration injection depicts a near-ideal square wave, it usually takes the uncorrected detector output 3 to 5 hours to reach a similar activity concentration (based on overlapping uncertainty bounds). Often, something that is more clearly evident in the deconvolved output for calibration injection events are times when the source capsule had not been adequately flushed prior to the start of an injection (see Figure 6c). In these cases, a clear "overshoot" of radon concentration is observed, followed by a gradual return to the target concentration over the 5-hour injection period as the excess radon is flushed from the detector's main delay volume. These tests of the deconvolution routine on extreme changes under controlled conditions provide confidence in the fidelity of the technique under real-world conditions.

## 4.2 Validation of best estimated radon activity concentration under real-world conditions

Changes in air mass fetch, transitioning from terrestrial to oceanic or vice versa, lead to significant variations in atmospheric constituent. When associated with strong frontal systems, such changes can occur over short timescales – comparable to the 3–5 hours it takes a two-filter radon detector to fully respond to a sustained calibration injection.

In Figure 7, we illustrate the response of $CH_4$ as measured by a fast-response analyzer (0.4 Hz) to a rapid change in air mass fetch during the winter period (22–23 February 2021) at HFD and WAO. Additionally, the Figure 7 presents the corresponding radon measurements, both with and without response time correction via deconvolution.

Initially, on February 22, a period of pollution is evident, characterized by elevated radon concentrations and $CH_4$ amount fractions, attributed to continental influences. However, a rapid transition to baseline conditions for both gases occurs as air masses shift towards cleaner oceanic origins on February 23. This transition is corroborated by air history maps generated using the Met Office NAME Lagrangian atmospheric dispersion model for each site (not shown here).

Analyzing the reference $CH_4$ time series reveals the elimination of time lag between $CH_4$ amount fraction and deconvolved radon compared to the initial measurements. Moreover, at both sites, the duration of $CH_4$ transition from polluted peak to baseline conditions aligns with deconvolved radon measurements.

Accurately representing radon concentrations holds significant importance, particularly when radon is used as a quantitative tracer such as improving and validating ATMs as well as estimating local-to-regional GHG fluxes via RTM (Biraud et al., 2000; Levin et al., 2021).

Trace gases can mix through the depth of the planetary boundary layer (PBL) on timescales of around 1-hour (Stull, 1988). Consequently, before attempting to employ radon to improve understanding of the dilution of atmospheric constituents with surface sources it is crucial that instrument response time is accounted for (see Figure 10, (Griffiths et al., 2016)). Here we further demonstrate the effectiveness of the deconvolution technique under real-world conditions by analysing diurnal composites of radon, $CH_4$, $CO_2$, and $N_2O$. The diurnal cycle of trace gases with surface sources is characterised by a morning maximum, when the mixing depth is shallowest, and an afternoon minimum, when convective mixing is deepest. After sunrise, when the nocturnal inversion breaks down and the daytime inversion begins to grow, concentrations reduce rapidly from their morning maximum to their afternoon minimum values. In the absence of short-term sinks (e.g., photosynthesis) or temporal changes in source function, all gases should be impacted equally by this dilution.





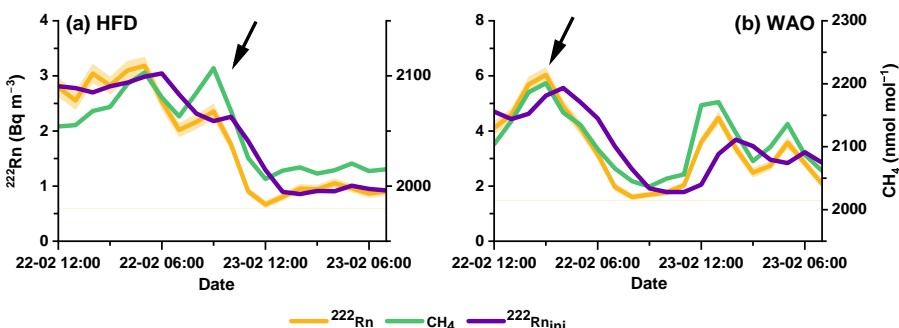

**Figure 7.** Response of $CH_4$ and radon (initial and best estimate) to rapid air mass changes in winter: a) HFD and b) WAO.

Focusing primarily on the warmer season - summer is crucial for our analysis because of gases pronounced diurnal variation in PBL height. This variation offers an ideal testing ground to evaluate the effectiveness of the radon deconvolution tool.

Hourly median summer diurnal composites of best estimated radon (deconvolved and STP corrected) and GHGs at TAC, HFD, and WAO are presented in Figure 8. At each of these sites, radon and the GHGs were sampled from the same (uppermost)
inlet height. However, there was an exception at TAC, where radon was sampled at 175 m a.g.l., while GHGs were sampled slightly higher at 185 m a.g.l.

During summer nights, the sampling inlets at TAC and HFD (175 m and 100 m a.g.l., respectively), are typically above the inversion layer. Consequently, median levels of trace gases for these sites remain relatively stable throughout the night. A noticeable increase begins around 05:00 LT (coinciding with sunrise) and peaks at 08:00 LT. Figure 8a (TAC and HFD)
indicate synchronous 08:00 LT peaks in radon and $CH_4$ at both towers. This behaviour is consistent with the destruction of the inversion layer at some time after 05:00 LT, allowing gases that had accumulated near the surface to mix upwards, past the sampling inlets (typically at or before 08:00 LT). By contrast, WAO measurements are consistently made well within the nocturnal inversion (10 m a.g.l.), and a distinct pattern of decreasing $CH_4$ and radon levels is observed after sunrise (WAO: Figure 8a).

The observation of simultaneous peaks in $CH_4$ and deconvolved radon across all three sites with contrasting sampling heights is a clear validation of the deconvolution technique, highlighting its ability to accurately represent the significant impact of diurnal atmospheric processes on radon concentrations.

The difference in radon concentration peaks compared to $CO_2$ and $N_2O$ amount fractions at all sites, however, highlights the unique response of these gases to both physical and biological processes (Figure 8b–c). The morning peak of $CO_2$ is observed
to occur three hours earlier than the radon peak at both TAC and HFD sites. This difference can be attributed to the onset of photosynthetic activity. The "switching off" of the photosynthetic sink at sunset, in combination with nighttime respiration by plants and soil organisms, and anthropogenic emissions, lead to a marked nocturnal $CO_2$ accumulation that is not observed in radon at HFD and TAC. This pattern is evidenced by Figure 9 which shows $CO_2$ measurements at the lowest sampling point of HFD (50 m a.g.l.), and Figure 8b showing WAO $CO_2$ amount fraction. The breakdown of the inversion layer around 05:00 LT
facilitates vertical mixing of accumulated $CO_2$ at 50 m — a process clearly depicted in the Figure 9. Additionally, the more





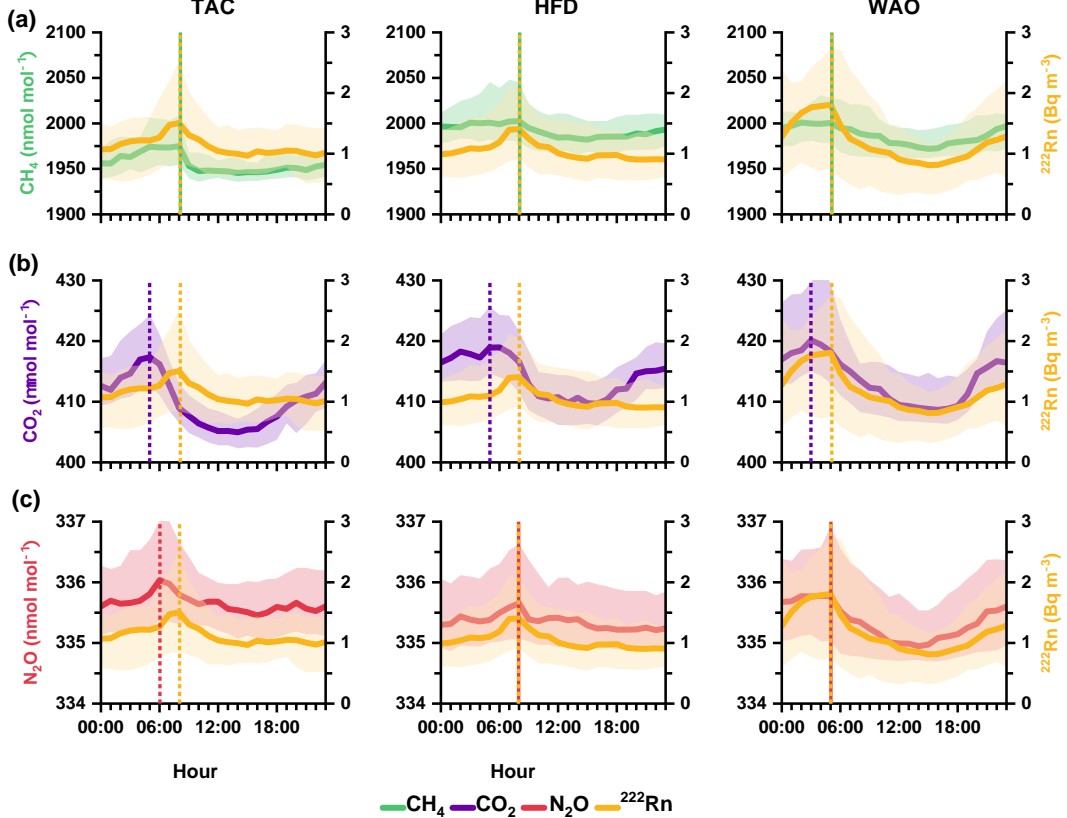

**Figure 8.** Summer diurnal composites of best estimated radon activity concentration and GHG amount fractions at TAC, HFD and WAO shown in panels: a) radon and $CH_4$, b) radon and $CO_2$, and c) radon and $N_2O$. Composites are based on hourly median values spanning June to August. Shaded areas indicate the 25th to 75th percentile to visualize variability of gases.

pronounced decrease in $CO_2$ compared to $CH_4$ can be attributed to photosynthetic activity that begins with sunlight; $CO_2$ is rapidly absorbed by vegetation, resulting in a notable reduction in atmospheric $CO_2$ levels.

At the WAO coastal site, the difference in median peak times between radon and $CO_2$ is likely driven by their distinct source/sink functions (Figure 8b). Radon exhibits a relatively constant source/sink function compared to $CO_2$. Consequently,
the radon concentration observed at 10 m a.g.l. is primarily influenced by atmospheric dynamics, particularly the formation and breakdown of the nocturnal inversion, with the peak typically occurring just before the inversion breaks down. Conversely, $CO_2$ experiences influences from both dynamics and a temporally variable source/sink function. At WAO, $CO_2$ peaks earlier than radon because photosynthesis begins immediately when the sun rises, drawing $CO_2$ out of the nocturnal inversion before convective mixing begins, which typically takes an hour or two to initiate due to the warming up process. However, the shaded
areas representing the 25th to 75th percentile range for $CO_2$ variability suggest that this peak frequently coincides with the radon peak.





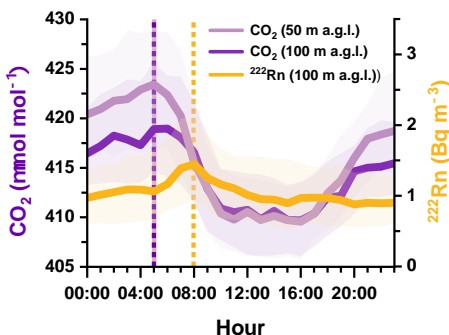

**Figure 9.** Diurnal composites of best estimated radon activity concentration and $CO_2$ amount fractions measured at 50 and 100 m a.g.l. at HFD during summer. Composites are based on hourly median values. Shaded areas indicate the 25th to 75th percentile to visualize variability of gases.

The peak for $N_2O$ is observed to coincide with the radon peak (Figure 8c), highlighting a pattern of simultaneous peaking for these gases at the site. This observation suggests that $N_2O$ lacks sunlight-sensitive sources/sinks, and its concentrations at this site and time of year are primarily influenced by atmospheric dynamics and a relatively stable surface source.

## 5 Recommendations

### 5.1 Instrument maintenance

As detailed in Section 2.2.2 some periodic maintenance of ANSTO two-filter radon detectors is required.

**Weekly:**

– Visually check that data are updating half-hourly to ensure continuous logging. This can be done remotely.

**Monthly:**

– Synchronise the data logger and PC clocks.

– Verify that scheduled events such as calibrations and backgrounds have occurred. This can also be done remotely.

**Quarterly:**

– If not using a compressed air calibration system, inspect and shake the calibration unit's desiccant tube. Confirm that the injection flow rate is within the range of 0.10–0.15 L m$^{-1}$.

**Yearly:**

– Check the detector and inlet system for leaks, with half-yearly checks recommended for detectors located outside.

– Inspect the plumbing of the calibration system.





– Check or replace the coarse aerosol filter on the inlet line.

**2-yearly:**

– If using a compressed air calibration system, replace the gas bottle.

**5-yearly:**

– Refresh the materials inside the detector head (stainless steel mesh filter and ZnS(Ag) sheet). If components of the aluminium frame of the head have started to corrode, the head casing should also be replaced. ANSTO can provide this
service or instructions for replacement (contact: radon@ansto.gov.au).

**10-yearly:**

– Replace all moving and electrical parts, including power supplies. ANSTO can assist with this service or provide instructions.

## 5.2   Instrument calibration and background

By default, the user specifies a schedule for instrument calibration and background checks. From a meteorological perspective, the calibration events will occur randomly. Each year, some calibrations will occur under conditions that closely satisfy the assumptions of the calibration process. Only these events should be retained.

Alternatively, background events can still be scheduled on an automatic, regular basis, but the calibrations can be remotely triggered every 1–2 months, according to forecast meteorological conditions better meeting the assumptions of the calibration
process. Automatic background checks will not account for any self-generation of radon inside the detector by trace amounts of $^{226}$Ra. To ensure continuity in case of software crashes or other issues, it is recommended to schedule background events to run automatically on a 2-monthly basis.

To achieve the lowest possible calibration uncertainty, a mobile calibration transfer standard device can be moved to the site once per year to run in parallel with the 1500 L radon detector for around two-weeks, to transfer a traceable calibration and
instrumental background. The transferred background will account for self-generation of radon inside the detector by trace amounts of $^{226}$Ra.

## 5.3   Software

For Windows 10 operating systems and higher the Visual Basic version of the RDM control software has been increasingly unstable. For these systems, or to operate the software from a Unix/Linux platform, we recommend moving to the Python based
version of RDM (https://github.com/anstoradonlab/radon-monitor/releases/). Additional features of the Python based control software include access to raw 10-second output of all detector parameters (as well as the 30-minute averages), access to a log of all system messages and error messages, access to a comprehensive log of all operational parameters of the calibration unit (if using a compressed air Burkert calibration system).





The graphical radon data processing software to perform all procedures described in Section 3.2 on the 30-minute instrument
output apart from the deconvolution process was developed (available upon request). The software was currently developed
and tested under Windows 10 or higher platforms.

## 5.4 Quality health workflow: three stages

A three stage data quality control regime is recommended:

**Monthly:** new data files should be checked for timestamping errors (missing, partial or duplicate records) and data flagged
if key diagnostic parameters are out of range (Table 1).

**Quarterly:** Monthly calibrations and quarterly background check should be processed and reviewed, and if feasible fitting
should be conducted.

**6-month (final data product):** 6-month fitted calibration coefficient and instrumental background are reviewed, along the
executing of deconvolution code and STP correction to derive the best estimate of radon concentration together with associated
uncertainties. In this stage, it is recommended that a simultaneous review of data from multiple sites to be conducted. This
approach enables for comprehensive analysis and comparison, facilitating the identification of any anomalies or inconsistencies
that may require further investigation.

## 5.5 Data levels for dissemination

Data obtained from ANSTO two-filter radon detectors can be disseminated across various temporal scales, including near
real-time (level 1) and 6-month (level 1) or long-term intervals. Each temporal scale is associated with different relative un-
certainties, which should be clearly documented in the metadata. To process raw counts into radon concentrations, aside from
response time correction, it is necessary to remove background counts and apply calibration. Both the background count and
the calibration coefficient can be modelled, with the accuracy and complexity of these models varying based on the available
data.

When a detector is first commissioned, initial estimates of calibration coefficient and instrumental background count are
determined. These values serve as constants for deriving near real-time radon concentrations during the first 6 months of
operation. Subsequently, the calibration and background history is reviewed, and fitted values for the next 6 months are derived.
To update the calibration coefficient and instrumental background for the subsequent month (e.g., month 7), the last value from
the fitting is utilized. It is noteworthy that the commissioning background check may involve thoron contamination and may
need to be disregarded. Regarding the deconvolution of near real-time data, a newly developed Rust-based "production code"
should be employed. This code typically takes up to one minute to deconvolve a day's worth of data. However, there is a
Python-based code "research code" available for the the level 1 data product.

The availability of level 1 data on 6-monthly basis is highly desirable for the modelers, particularly for evaluating operational
data model integration, in this case employing radon to assess the accuracy of ATMs. Implementing a routine and operational
radon approach facilitates the ongoing assessment of the accuracy of these transport models, thereby enhancing the accuracy
of GHG emission estimates.



After one full year of measurements are available, the consistency and seasonality of calibration factors can be assessed, and a representative average value calculated for the year. The available background checks can be assessed for consistency and a linear model applied. This year of data should be processed with this information and archived, and this calibration factor and background model should then be used to generate the "near real time" data for the subsequent year of measurements.

This process is complete after 5 years when the detector's measurement head is replaced, and the process starts again.

## 6 Conclusions

In this study, we have developed a comprehensive protocol to enhance the reliability of reported radon activity concentrations for real-time applications, enabling their direct comparison alongside GHG measurements within a monitoring network of three independently managed observatories in the UK.

Our protocol emphasizes the importance of achieving radon measurements with precision and resolution comparable to GHGs, necessitating specific procedures for quality control, calibration, deconvolution, and uncertainty assignment. Validation of this protocol through meticulous analysis of calibration events and comparison with GHG measurements establishes a foundation for standardizing protocols across networks (ICOS and GAW) using two-filter instruments to measure radon.

The critical role of deconvolution in preserving the true signal of radon, particularly for quantitative tracer applications in atmospheric studies, is underscored. Our analysis demonstrates the adaptability of the deconvolution technique across seasonal variations and distinct atmospheric dynamics, ensuring accuracy in tracking GHG trends across all sites and validating ATM outputs. Synchronization of radon peaks with GHGs, especially during summer diurnal variations, further validates the effectiveness of our deconvolution method in real-world conditions, highlighting radon's importance as a tracer in understanding atmospheric mixing.

This study offers a robust protocol for radon measurement within GHG monitoring networks and underscores the invaluable role of radon as an independent metric in ATMs. By ensuring the accuracy and comparability of radon measurements, may contribute to refining GHG emissions estimates and improving understanding of atmospheric processes, supporting global efforts to mitigate climate change in alignment with the Paris Agreement goals.



*Code and data availability.*  The graphical radon data processing software described in Section 5.3 is available upon request. The runtime script for the python-based research package used for this study is available upon request. GHG data from the Heathfield, Tacolneston and Weybourne Observatory are available from the Centre for Environmental Data Analysis (CEDA) data archive:

– https://catalogue.ceda.ac.uk/uuid/df502fe4715c4177ab5e4e367a99316b,

– https://catalogue.ceda.ac.uk/uuid/ae483e02e5c345c59c2b72ac46574103, and

– https://catalogue.ceda.ac.uk/uuid/36517548500e1e4e85c97d99457e268a.

Radon data presented in this paper are available upon request.

## Appendix A:  Appendix A

**Table A1.** Site-specific radon source strength (Pylon 2000A $^{226}$Ra).

| Site | Source strength (kBq) |
| --- | --- |
| TAC | 49.138 |
| RGL | 49.197 |
| HFD | 49.311 |
| WAO | 41.822 |



**Table A2.** Summary of dual-flow-loop two filter radon detector outputs, description and units

| Detector output (parameters) | Description | Unit |
|---|---|---|
| Date | timestamp represents the end of measurement interval | dd mm yy hh:mm:ss |
| external_flow | moves sampled air from intake point, inlet pipe, thoron delay, primary filter, into the detector delay volume and out of exhaust valve | L min$^{-1}$ |
| internal_flow | Circulates sampled air from the enclosure containing both blowers, through the flow homogenising denim screen, down the length of the delay volume, through the detector head and secondary filter, along the central pipe, and back to the blower enclosure | m s$^{-1}$ |
| LLD | lower level of discrimination | count (30 min)$^{-1}$ |
| ULD | upper level of discrimination | count (30 min)$^{-1}$ |
| differential_press | differential pressure between the detector tank and ambient air | mV |
| logger_temp | temperature inside the logger | $^\circ$C |
| detector_temp | temperature inside the detector tank | $^\circ$C |
| detector_RH | relative humidity inside the detector tank | % |
| detector_press | absolute pressure inside the detector tank | hPa |
| voltage | photomultiplier high voltage | V |
| bg_lld | instrumental background | count (30 min)$^{-1}$ |
| cal_coeff | monthly calibration coefficient | count s$^{-1}$ (Bq m$^{-3}$)$^{-1}$ |
| bg_lld_inter | 6-month interpolated background | count (30 min)$^{-1}$ |
| cal_coeff_inter | 6-month interpolated calibration | count s$^{-1}$ (Bq m$^{-3}$)$^{-1}$ |
| radon_initial | calibrated radon activity concentration based on LLD, calibration and background interpolated value | Bq m$^{-3}$ |
| radon | deconvolved calibrated radon activity concentration corrected for standard temperature and pressure (best estimate of radon activity concentration) | Bq m$^{-3}$ (STP) |
| radon_uncertainty | combined uncertainties | Bq m$^{-3}$ (STP) |



*Author contributions.* DK led the research design, performed all data visualizations, and developed the concepts and results. The development of a standardized protocol was a joint effort by DK, EC, SC, AG, TA, GF, and AW. Radon measurements and calibration at HFD were

conducted by DK, at WAO by GF, and at TAC by AW. EC developed the software for radon quality control, performing data checks and deconvolution run across all sites, with support from DK, AW, GF, and AG. GHG measurement responsibilities were divided as follows: TA and CR for HFD, GF, LF, PP, KA for WAO, and SD, JP, DY, KS and AW for TAC. DK wrote the complete original draft of the paper, with SC, AG, PP, AW, EC, JP, CR and TA contributing to the editing and review process.

*Competing interests.* At least one of the (co-)authors is a member of the editorial board of Atmospheric Measurement Techniques.

*Acknowledgements.* Funding for this work is primarily through the NPL Directors' Fund, National Measurement System Funding and NERC - Building a Green Future: GEMMA Greenhouse Gas Emissions Measurement and Modelling Advancement project. The ANSTO radon detectors for this work and the Picarro G5310 at HFD were purchased from the UKRI NERC grant 'Advanced UK Observing Network For Air Quality, Public Heath and Greenhouse Gas Research' in 2018, awarded to University of Edinburgh, University of Bristol and University of East Anglia. The authors thank Andrew Manning and Alex Etchells (both University of East Anglia) for maintained with

support $CO_2$ measurement system at WAO and Mr Ot Sisoutham of ANSTO for construction and initial testing of all two-filter radon detectors used in this study.





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
