# Peer review of "Direct high-precision radon quantification for interpreting high frequency greenhouse gas measurements"

_Atmospheric Measurement Techniques, 2024_

## Author Comment (AC1)

**Dafina Kikaj, Senior Scientist**
**National Physical Laboratory**
**Teddington, UK**

**15th October 2024**

**Dr David Griffith, Editor**
**Atmospheric Measurement Techniques**

Dear Dr Griffith,

Please find below our response to the comments from the **anonymous referee #1** on our paper: **"Direct high-precision radon quantification for interpreting high frequency greenhouse gas measurements"**.

We would like to thank the reviewer for taking the time to look over our manuscript and provide us with comprehensive constructive feedback.

Best regards,

Dafina Kikaj
(on behalf of all co-authors)

**Response to Anonymous reviewer's 1 comments**

*In this article, Kikaj et al. present in great detail the workflow to operate ANSTO dual-flow-loop two-filter radon detectors, and to process data to achieve best-quality radon observation suitable for atmospheric transport model validation and greenhouse gas (GHG) emission estimation (among many other goals, not necessarily mentioned by the authors). The authors finally provide recommendations that could be used to establish standard operation procedures in international cooperation programmes such as ICOS or GAW.*

*General appreciation*

*This article will be of undeniable value for the growing scientific community operating two-filter radon detectors across the world, and for the dissemination of high-quality radon data for various scientific or operational uses. The article is fully in the scope of Atmospheric Measurement Techniques and meets the required quality standards of the journal, both in terms of scientific robustness and presentation quality.*

*Therefore, I recommend the publication of this manuscript. Before this, minor revisions could be considered. Especially, some aspects pointed in my comments make the article sometimes hard to understand and deserve to be clarified. Numerical values of instrument- or site-specific parameters are also sometimes missing.*
**Thank you very much for your positive evaluation of our manuscript. We have carefully addressed your specific comments in the sections below. We appreciate your time and feedback.**

*Specific comments*

*Line 20: « to keep this ... »→ « to keep the ... » ?*
**Thanks for the comment. We chose to retain the original text.**

*Lines 20-55: This is a long (but nevertheless interesting!) introduction to justify the need of high-quality radon measurements. But this need would still be evident even introduced more briefly. So, was the manuscript to be shortened, could these paragraphs be reduced. Beyond the validation of ATMs and the estimation of GHG*

*emissions, radon data have many other interesting application fields in atmospheric sciences. I miss a bit of a wider view on applications while reading this introduction (and the final conclusions).*

**We have shortened this text a little but would prefer to retain most of the detail, since it represents the main purposes for which the UK radon measurement network as so far been used. We have added a brief summary of other radon measurement applications near the end of this section in the revised manuscript.**

*Line 141: Does « local time » refer to the official time in the UK (including winter/summer changes), or the local solar time (i.e. UTC)? The latter is much more relevant when considering diurnal cycles in the planetary boundary layer. Please specify.*
*The following change has been made in the revised text:*
**"… and reported in Local Solar Time (LST; equivalent to UTC in these locations)."**

*Line 163: Is WAO a global or regional GAW site?*
**WAO is a regional GAW site. This has been corrected in the revised text.**

**Line 232:** What is the empirical procedure to determine an appropriate flushing time? What are the criteria? Please go into more detail.
*We have added a more detailed in the revised manuscript:*
**"Prior to initiating a calibration, the $^{226}$Ra source should be well flushed, with the exhaust directed to the outside ambient air at a point well removed from the detector sampling location. The optimal flushing time for each calibration setup is assessed by the shape of the calibration curve. The curve should rise monotonically to a near-constant concentration after ~5 hours. If the curve rises steeply, overshoots, and then begins to decrease again, the source was not sufficiently flushed. The flushing time should be determined empirically for each installation and will depend on factors such as the source strength, the flushing flow rate, and the time since the source was last flushed. The flushing flow rate is usually lower (0.8–0.15 L min$^{-1}$) if the source is being flushed with ambient air (to minimise the amount of ambient $^{222}$Rn and $^{220}$Rn introduced to the system) but can be higher (0.15–0.25 L min$^{-1}$) if being flushed with dry, radon-free air (to improve the consistency of radon delivery). A flushing period between 5–10 hours is usually sufficient. Once the initial flushing is complete, a calibration is then performed by continuing to pass air through the source, but this time directed into the detector's sampling airstream, for a period …"**

**Figure 2:** It seems that this figure uses data from a real calibration. Nevertheless, the linear trend estimated from outdoor ambient air before and after the calibration is hardly visible. For clarity, could a schematic be presented instead of real data? Further, I suggest that the variable names LLDpeak and LLDpeak,a should be displayed in the figure or in the legend. Finally, the figure should be enlarged (legend characters too small).
**Thank you for your suggestion. Figure 2 was generated using data from the January 2018 calibration event at the Cabauw ICOS research tower, where the sampling inlet height is 20 m above ground level (a.g.l.). Since the source in this case is smaller, it highlights the linear trend in the estimated outdoor ambient concentration more clearly. We have also added the labels for clarity.**

**Lines 251-252:** « This scaling factor will vary with the length of the calibration injection and sampling flow rate. »: What are the values retained for the 3 stations? More generally, what is the procedure to determine the scaling factor?
**In the submitted manuscript, we have already referenced the Griffiths et al. (2016) paper, where detailed information can be found. The scaling factor is consistent across all three sites, with the primary variation depending on the sampling flow rate, which differs for detectors of varying sizes (e.g. 1500 L vs 700 L). We have provided additional details in the revised manuscript as follows:**

*Original text:*
**"This scaling factor will vary with the length of calibration injection and sampling flow rate."**
*Revised text:*

**"A Python notebook that provides an analytical solution to the model equations and can be used to calculate the scaling factor is available at https://github.com/anstoradonlab/radcalsim."**

**Line 243:** The full-text development of the acronyms LLD and ULD is too discreetly hidden in Table A2. Please define these important variables in the main text body.

*The following change has been made in the revised text:*

**"Output from the photomultiplier is amplified and fed into a discriminator. Total counts above a threshold of 0.5 V are recorded as LLD (lower level of discrimination) and total counts above a threshold of 1.0 V are recorded as ULD (upper level of discrimination). It has been empirically determined that ULD counts essentially represent different forms of noise, so it is the number of counts that lie within the 0.5–1.0 V pulse-height window that are used to calculate the radon activity concentration (in Bq m$^{-3}$). When ULD counts are low, valid counts are determined as LLD – ULD. However, when ULD counts are high, that observation period should be flagged as bad data. The timestamp associated with a count represents the end of the measurement period."**

**Lines 279-280:** It is not clear why a high calibration accuracy is needed to derive radon vertical gradients from tall towers. Could you provide values to support this statement (e.g. typical radon vertical gradients over 100 m vs. typical measurement accuracy)?

**Explanation for the reviewer:**

During the day, when mixing in the atmospheric boundary layer is strongest, radon gradients over the lowest 50 to 100 m of the atmosphere can be as low as 0.1–0.2 Bq m$^{-3}$ (e.g., Chambers et al. 2011, 50 m tower in Sydney, Australia; but also **A. Frumau, pers. comm., from the 213 m a.g.l. research tower in Cabauw ICOS research tower, the Netherlands**, and **I. Levin, pers. comm., from the 100 m a.g.l. research tower at Karlsruhe, Germany**). The response time of ANSTO two filter radon detectors is two slow for a single detector to be multiplexed for multi-height (gradient or profile) measurements. This means that separate (independently calibrated) detectors are required to measure concentrations at each height. Daytime outdoor radon concentrations in the surface layer are typically of order 1–3 Bq m$^{-3}$, and for radon concentrations >1 Bq m$^{-3}$, the measurement uncertainty of ANSTO 1500 L detectors is < 10% (see Grossi et al. 2020).

*We have added an explanation to this effect at the end of **Line 282** in the revised version of the manuscript as follow:*

**"Regarding gradient measurements, during the day, when mixing in the atmospheric boundary layer is strongest, radon gradients over the lowest 50 to 100 m of the atmosphere can be as low as 0.1–0.2 Bq m$^{-3}$ (e.g., Chambers et al. 2011). The response time of ANSTO two filter radon detectors is two slow for a single detector to be multiplexed for multi-height (gradient or profile) measurements. This means that separate (independently calibrated) detectors are required to measure concentrations at each height. Since daytime radon concentrations over land in the surface layer are typically 1–3 Bq m$^{-3}$, and the measurement uncertainty of ANSTO 1500 L detectors is < 10% for radon concentrations > 1 Bq m$^{-3}$ (see Grossi et al. 2020), it is usually possible to reliably determine such gradients when detectors are appropriately calibrated. "**

**Lines 301-302:** « the instrument undergoes few measurement cycles ... » : What is here meant as a « cycle » ? A 30-min count sequence? I am afraid this term could be a bit confusing here, please rephrase.

*The following change has been made in the revised text:*

**"… the instrument then undergoes several 30-minute measurement cycles (1-2 hours) to readjust itself and return to reporting ambient levels correctly."**

**Lines 318-319:** The inlet heights could be provided in Sections 2.1.1 and 2.1.2.

**Actually, we appear to have duplicated this information. It is already mentioned in Sections 2.1.1 and 2.1.2. We will remove mention of it here in the revised manuscript to avoid repetition.**

**Lines 344-345:** « Totalized counts include (…) and sample flow rate »: this sounds strange to me. LLD and ULD are clearly totalized counts, but how can it be so for a flow rate? Do you mean that the measured flow rate is integrated over 30 min and recorded as a total air volume? Please clarify.
*The following change has been made in the revised text:*
**"… and sample flow rate (measured using a domestic gas meter that has a 2.0 L cyclic volume with individual cycles counted optically), while all other parameters are averaged (see Figure 3)."**

**Figure 3:** There is a legend concerning values in red, green and yellow, but nothing is said about the parameters in bold black.
**Thank you for noticing this omission. It has been corrected in the revised version of the manuscript.**

**Lines 361-362:** « a flow rate of 5.5 m s-1 » : Usually flow rates are expressed as volumes (or masses) per unit time. Why is it expressed as a velocity here? What is the section area needed to convert this air velocity into m³ s-1?
**Explanation:**
**We acknowledge that we mistakenly expressed the internal flow rate in m³ s⁻¹ in some instances (e.g., Figure 3). We recognize the confusion regarding the use of velocity to describe flow rate. In this case, the value "5.5 m/s" refers to the** linear flow rate **(velocity) rather than the volumetric flow rate.**

*The following change has been made in the revised text:*
**"The flow rate of the internal flow loop should be sufficient to exchange all the air within the detector through the measurement head in less than 3 minutes (the half-life of $^{218}$Po). Flow within the central pipe of the detector is measured as a velocity (V) with an insertion probe at the centre of the 50 mm ID pipe. Based on the typical velocity profile in a pipe, the actual flow rate is estimated as 80% of the maximum flow rate $Q_{max}$. While a flow rate of around 5.5 m s⁻¹ is technically sufficient for this purpose, a faster rate is desirable, and values of 6–12 m s⁻¹ are typically achievable (based on individual blower performance and flow impedance of the measurement head)."**
**Addition:** ($Q_{max} = 6x10^4 \ p \ (0.025^2)$ (L.min⁻¹)).

**Line 376:** What is a UPS? Please write this acronym in full text.
*The following change has been made in the revised text:*
**"… through an uninterruptable power supply (UPS) …"**

**Line 396:** « if the blowers restarted early »: do you mean that the blowers may unexpectedly restart earlier than planned? This sentence sounds strange to me.
*The following change has been made in the revised text:*
**"… the last 30-minute sample was excluded if the blowers restarted early (due to the small-time differences that can happen between the PC clock and the logger clock if they are not regularly synchronised), and a check was made …"**

**Lines 396-397:** « a check was made that the remaining data were linear (approximately constant) with relatively low variance »: Background counts are expected to show no trend within the retained measurement sequence, but to remain constant (only subject to count noise). The adjective « linear » is here confusing. Further, the count variance should be low relative to what ? What is the retained value for a variance threshold?
**Apologies for this confusion. The use of linear here was redundant and will be replaced with "approximately constant". Here the term "low variance" was meant to indicate no "noise" counts ("spikes"). If no noise counts were present during the background period, no specific threshold for variance was applied.**
*The following change has been made in the revised text:*
**"… a check was made that the remaining data were approximately constant and free of noise counts."**

**Lines 407-408:** « a site-specific threshold value » : Is the threshold value of 50 counts min$^{-1}$ valid for the three sites? It would be useful if the authors could provide a method to determine such a threshold at any site.

**We set the threshold value based on "expert judgment" after reviewing at least one year worth of data. This means the value is site-specific.**

*However, we suggest the following:*

**"Typically for coastal or island sites, the 10$^{th}$ percentile radon concentration is indicative of "baseline" (extended oceanic fetch) conditions. At other sites, a necessary criterion to keep a calibration event could be that for the 10 hours prior and 10 hours following the calibration event should be consistently less than the 20$^{th}$ percentile (2x baseline) monthly radon concentration. Since the atmosphere usually changes to or from true baseline conditions over periods of several hours or more (not from one 30-minute sample to the next), then an indication of near consistent baseline before and after a 10-hour calibration event is a reasonable indication that near baseline conditions persisted over the intervening period."**

**Section 3.3:** I must confess I had a hard time while reading this subsection and am still not able to reproduce the fitting algorithm based on the details give here. With a concern of reproducibility, I think that the algorithm should explained in more detail and clarity, possibly as supplemental material. Alternatively, a well-documented routine (in python or other freeware language) could be made available on a public repository. More specifically:

**We apologize for the confusion. We have now provided the implemented code as supplementary material (Code and Data availability section), which should enhance clarity and allow for better reproducibility of the fitting algorithm.**

*The following change has been made in the revised text:*

**"For the purpose of reproducibility, the implemented code is provided as supplementary material."**

**Lines 421 and ff.**: Could a literature reference be provided on the Savitzky-Golay method? As far as I know, this smoothing method consists of fitting a nth-degree polynomial on the data contained in a moving time window centred around the current timestep. The result is a single value equal to the value of the polynomial at this timestamp (with n=0, the method is thus a simple moving averaging). The whole process is done again considering a new time window centered around the next timestamp, and so forth. For this reason, the number of data points in the resulting smoothed signal is equal to the number of raw data points. But in the present study an interpolated curve is obviously produced between the timestamps of the raw data points, which requires to handle several polynomials produced within adjacent time windows. It seems that information elements on this aspect are given in Lines 432-434, but I do not understand those elements, sorry. What is done in steps 3-5 of the procedure is also unclear to me. All this should be clarified, or at least appropriate references should be given.

**The general outline you have laid out is correct. For our analysis, we loop through all 30-minute data points, regardless of whether they contain valid measurement data. The interpolated 'assigned values' are introduced to ensure that there are at least two data points available for the fitting step.**

**Steps 3 and 4 involve smoothing the curves because, during certain periods, there will be identical input values. As a result, the output from Step 2 may consist of a series of straight lines with abrupt changes.**

**Steps 5 is to handle the edges of data where there is only one data point, i.e. not enough data points within the moving time window.**

**As mentioned above, the implemented code is provided as supplementary material.**

**Lines 423-424**: After some effort I eventually understood that a 184-day (~ 6 months) moving window centred on the current timestamp is used. If I am right, couldn't you express this more simply than « two half-regression windows, each with a duration of 92 days » ?

*The following change has been made in the revised text:*

- **"… Calibration and background values are selected within a specific range. This range as 184-day moving window centred on each of the timestamps."**

- "…The result of the moving fit requires further smoothing. Centre moving average is used as 184-day moving window centred on each of the timestamps."
- "…The centre moving average is performed on the results of Step 2. This is to smooth out the effect of multiple adjacent data points sharing the same measurements for the fitting, which may result in step changes"

Line 427 : Why does the result of the moving fit requires further smoothing ? This does make no sense to me since the result of the moving fit is a linear curve (1st order polynomial). Please clarify.

**This is because the moving fit is done on data with gaps, which means that we have sections of data where within each section a same set of measurements are used for the fit, i.e. the chances of a measurements included in the fitting window for one timestamp being the same as the one for the next timestamp is going to be high. Without the smoothing, our series of linear fits will result in a series of straight lines, which we decided was not ideal.**

Line 431 : Does « supplemented » should be understood as « extrapolated » ?
*The following change has been made in the revised text:*
**"To ensure a smooth transition at the beginning of the dataset, the data points prior to the first calibration/background measurement will be assigned the fitted value from that measurement. Similarly, for the end of the dataset, the fitted value from the last calibration/background measurement will be used to fill in the remaining data points."**

Lines 442-442 : This is sophisticated way to express more simply that missing points are linearly interpolated between the closest available data points, and spaced evenly every gapmax days. Did I miss something more subtle ? Beyond this, what is the value of gapmax in your study ?

**Thank you for the advice, based on which we have made it simpler. However, we would like to make clear that the gap between the new points is likely to be smaller than gap_max.**
*The following change has been made in the revised text:*
**"The gap is then linearly interpolated between the closest available data points to construct this number of new data points, evenly spaced across the gap to the nearest 30 minutes. For Heathfield, gap$_\text{max}$ is set to 93 days"**

Lines 444-445 and ff. : Annual reductions in sensitivity are expressed as percentages. How are these reduction percentages defined from the fitted calibration coefficients ? As the reduction of the value relative to the preceeding year ? Please specify.

**Thank you for your question. The annual reduction percentages in sensitivity come from the fitted calibration coefficients and are calculated based on the previous year's values. It seems that the fitting procedure can make the annual rate of decline look larger compared to a linear model of calibration sensitivity change. This effect is sometimes worse in cases like Figure 5d, where edge effects play a role.**
*The following change has been made in the revised text:*
**"…varying from 3.2 % to 4.7 %. These reduction percentages are calculated based on the fitted calibration coefficients and represent the decline relative to the preceding year." And**
**"…specifically the ZnS(Ag) scintillation material. However, it is also worth noting that the fitting procedure can make the annual rate of decline appear larger compared to a linear model of calibration sensitivity change. This effect is sometimes more pronounced in certain cases, such as in Figure 5d, where edge effects come into play."**

Section 3.4 : The signal deconvolution to correct the slow instrument response (compared e.g. to GHG analyzers) is major aspect of the data processing. At first glance the algorithm is described in some detail in Griffiths et al. 2016, but I found no mention in this article, neither in the present manuscript, of code availability. Would routines be accessible somewhere ? This would be a considerable gain of time and effort for users.

**We have now provided the implemented code as supplementary material (Code and Data availability section). Software implementing the deconvolution algorithm can be found at https://github.com/anstoradonlab/rdfix**

Line 468, « the median of the deconvolution result at that timestamp » : this suggests the deconvolution result at a given timestamp is not a single radon volumic activity value. Further in the text (line 509), it is indeed written that the deconvolution process results in a statistical distribution. This is an important - but not straightforward! - point that should be clarified as early as here in the text.

**Apologize for this confusion. We are using mean and not median. We have revised the text in manuscript.**

Lines 501-502, « The maximum discrepancy (…) of 7.7%. » : This sentence is unclear. Should one understand that the discrepancy is of 7.7%? And 7.7% of what relative to what ? Please clarify.

**Thank you. To clarify, we conducted a calculation to compare the differences in water vapor between dry air (0.8879) and wet air (0.9558) for the highest pressure and temperature. This difference is expressed in percentage. We have revised the manuscript to improve clarity and ensure this point is clearly articulated.**
*The following change has been made in the revised text:*
**"To assess the necessity for correcting water vapor, a simple calculation was carried out to compare the difference between dry (fdry) and wet (fwet) air across a spectrum of UK extreme climate conditions, as indicated in the Table 2. The maximum observed discrepancy of 7.1 % indicated the difference in water vapor concentration between dry and wet air in scenarios of highest temperature and pressure."**

Section 3.7 : The whole subsection deserves to be clarified. It is entitled « Combined measurement uncertainty », and lists and quantifies (as induced relative uncertainties) different specific source of uncertainty. But in the end, no numerical value is given for the resulting combined uncertainty. It is nevertheless stated in the very first sentence that the [combined] uncertainty is eventually derived from the distribution resulting from the deconvolution process (as the difference between the 16th and 84th percentiles). That the different uncertainty sources listed in this section affect this distribution is far from being evident to most readers, I am afraid. Could it be explained why? Are the listed relative uncertainties the specific impact of each source on this final percentile difference ? How are these values estimated ? It could also be clarified that the combined uncertainty is here not obtained as a rooted sum of squares of individual uncertainties, as more usually done.

**The variables listed are the inputs for the deconvolution as described in Griffith's et al (2016), as mentioned in Line 127. The deconvolution is an inversion model, where variables are sampled within the given distributions to estimate the true atmospheric values that will result in our observed values. This means that the uncertainty is not calculated analytically, but numerically.**

Line 511, spelling : « Poisson ». Please specify how an uncertainty value can be derived from the Poisson distribution.

**Thanks for this, corrected. The deconvolution script will sample from the Poisson distribution.**

Lines 517-518 (« plate-out of unattached radon progeny ») and 522 (« plate-out effect ») : The term « plate-out » is obscure to me (it seems it has to do with particle deposition), and I don't know exactly what the plate-out effect is. Could a short explanation and/or a reference be given?

**"Plate-out" refers to the removal of radon progeny (e.g. 214-Po and 218-Po) through dry deposition to internal surfaces of the detector (e.g. the walls, pipes and wires). Since ambient aerosols are removed from the airstream as it enters the detector, the radon progeny (when they form) mostly remain unattached, charged and highly mobile, and are therefore susceptible to plate-out loss. This loss can be minimised if**

**near-laminar "plug-like" flow is maintained inside the main detector volume, which is the purpose of the detector's denim screen flow homogeniser. Empirical results indicate that the relative amount of progeny lost to plate-out effects is constant (to within a small uncertainty). The detector's calibration process takes this into account.**

Line 539, typo : 8th → 84$^{th}$
**Corrected.**

Section 4 : This section is very convincing about the capability of the deconvolution process to account for sharp changes in ambient radon with the right timing (by comparing the radon signal to GES concentrations obtained with fast analyzers), as well as for absolute values of radon volumic activity in controlled conditions. However, it could be thought that the deconvolution algorithm had already undergone careful validation in Griffiths et al. 2016. Could you specify what is new concerning the validation checks made in the present study compared to the original paper ?

**Although the deconvolution method has indeed been validated (Griffiths et al., 2016) the technique has not yet been widely used and has not been validated at the HFD and WAO sites. The algorithm is relatively complex and depends on the radon detector's behaviour closely matching the theoretical model, something which is not guaranteed. For these reasons, it is worth briefly demonstrating that the algorithm has been correctly implemented at this site. This section is also important for demonstrating that the response-time correction is useful for detecting unexpected non-constant radon delivery (Fig. 6c) which would otherwise bias the estimated calibration coefficient.**

*The following change has been made in the revised text:*

**"The purpose of the deconvolution algorithm is to correct for the delay between the sampled radon concentration and the detected signal, and here we demonstrate that the implementation of the deconvolution algorithm is successful at the HFD and WAO sites."**

Lines 557-558 : This sentence would be clearer if « (222Rnini) » was moved just after « radon concentrations ».
**Done.**

Figure 6 : The panels in this figure are too small. They could be enlarged in order to fill the full text width. A thin vertical dashed line could be added at 9:00 LT, when the signal reaches 85 % of the target value. It also seems there is confusion in colors between the figure and the text (lines 560-562). Especially, the deconvolved signal is not purple (cf. line 562) but obviously yellow, and vice versa. Also, the target square waves could be displayed in the graphs for direct comparison with the deconvolved signal.

**Thank you for this. We have made the following revisions:**

- **The vertical dashed line is added at 9 LST.**
- **We have corrected the colour discrepancy in the text.**
- **We have included the target square waves in the graphs for the direct comparison.**

Line 594 : Change to read « (see Figure 10 in Griffiths et al. 2016) ». If the manuscript has been prepared with Latex, please consider using \citep[][]{} to avoid double parentheses (here and at some other places in the text), e.g. in this case : \citep[see Figure 10 in ][]{Griffiths2016}.

**Thanks a lot for this. I have ensured that double parentheses are avoided throughout the manuscript by using the \citep[][]{} command in LaTeX, including in this case. This has now been corrected.**

Line 601 : probably missing dash after « summer ».
**Thanks, corrected.**

Lines 619-620 : If during the night the measurements at those sites are decoupled from the surface (the inlets being above the inversion), how the CO2 exhalated by the soil and accumulated near the surface could reach the inlets 3h before the radon, which also accumulated in similar conditions below the inversion? I am not convinced by the explanation given here for this 3h delay. Should an additional source of CO2 be hypothesized?

**Thank you for your insightful comment. Upon further reflection, we realize that we made an error in our initial explanation. We have been using the "seasonal median" (composites made based on hourly median values) for our analysis, which would remove the most extreme, strongly stable nighttime conditions. As a result, during these periods, the inlet height would actually be located just at the top of the inversion layer (but still inside).**

**Given this, the earlier detection of $CO_2$ at the inlets can be explained by the fact that, under these conditions, the air at the top of the inversion is still connected to the surface during the night. The $CO_2$ exhaled by the soil can therefore rise and reach the inlets more rapidly than radon, which tends to accumulate and mix upward more slowly from below the inversion.**

**The observed 3-hour delay between the $CO_2$ and radon signals could also be attributed to differences in the transport dynamics of these gases. $CO_2$, being continuously released by soil respiration, is more quickly affected by surface air movements and sunlight-triggered processes like photosynthesis. Radon, on the other hand, is less influenced by these immediate factors and thus takes longer to mix upward from its accumulation near the surface.**

**Therefore, we do not believe an additional source of $CO_2$ needs to be hypothesized. Instead, the difference in how these gases mix and move through the atmosphere under the influence of the inversion layer and surface air exchange appears to account for the delay.**

**We clarified this explanation in the manuscript and correct our oversight regarding the use of the seasonal median.**

Figure 8 : The interquartile range for radon is displayed in light yellow and is hardly visible. Despite the use of color transparency, the superposition of variability strips is confusing. The variabilty of radon could be alternatively displayed e.g. with thin dashed curves.

**Thank you for this. We made an effort to adjust the colour slightly to improve visibility, but we still prefer to present it as a shaded area for the sake of consistency throughout the figures.**

Line 628-631 : For this coastal site nothing is said about the possible influence of land/see breezes. Could breezes play a role in the radon and GHG diurnal cycles ?

**Thank you for your comment. We agree that breezes likely influence the radon and GHG diurnal cycles in this coastal site and have added a sentence in the manuscript to acknowledge this. However, we opted not to explore the land/sea breeze effects further, as it falls outside the primary scope of this paper. Our focus is on the broader mechanisms driving the diurnal cycles of radon and GHG, particularly as GHG are used as validation for the deconvolution process. While breezes are important, a detailed analysis of their influence would extend beyond our intended scope.**

Figure 9 : the CO2 curve at 50 m agl could be integrated to the central panel in Figure 8, and in turn Figure 9 could be removed.

**Thank you for your feedback regarding the figures. While we appreciate your suggestion to integrate the $CO_2$ curve at 50 m a.g.l. into the central panel of Figure 8, we prefer to maintain this as two separate figures for consistency. We believe that retaining Figure 9 will provide valuable context and clarity for the data presented.**

Section 5 : The recommendation summarized in this section will be very helpful for operators of ANSTO radon detectors, thank you for them!

**Thank you for your kind words regarding the recommendations provided in this section. Your feedback is greatly appreciated and reinforces the importance of our work.**

Line 687 : « three stages » is not useful in this subsection title.

**We removed it.**

Lines 710-712 : It is not clear what is the difference, in term of performance, between the Rust- and Python-based codes. It sounds like if the Python-code is less efficient or gives poorer results. Is it the case ? Could you clarify ?

**The Rust-based implementation of the deconvolution algorithm produces results which are similar to the original Python-based code. Its advantages are mostly related to usability: (1) the installation size is < 10Mb vs 2.6Gb, (2) the user is presented with a well-defined command-line interface rather than having to write a Python script, (3) a Windows binary is available in addition to a Linux binary, compared with the Linux-only Python version, (4) it is a simple one-command installation, and (5) the Rust version executes more quickly (but this is not very important, it does not execute much more quickly).**

**Considering it now, the distinction made in this paragraph between the two version of the deconvolution code is not particularly helpful, and we will remove the mention of the two implementations of the deconvolution code so that**
*Original text:*
**"Regarding the deconvolution of near real-time data, a newly developed Rust-based "production code" should be employed. This code typically takes up to one minute to deconvolve a day's worth of data. However, there is a Python-based code "research code" available for the the level 1 data product."**
*Will be changed to*
**"For near real-time measurements, deconvolution can be run routinely with the code taking about one minute to process one day's worth of data. Deconvolution is then re-run in postprocessing after the finalisation of the calibration coefficients and background count rate timeseries."**

Line 712, typo : « … the the … »

**Thanks.**

Line 739 : That this measurement protocol could be a significant contribution to climate change mitigation and the achievement of the Paris Agreement is a far reaching conclusion! This nice and very useful article would not suffer to have its final point after « processes ».

I again thank the authors for their work.

**Thank for your time and valuable comments!**

---

## Author Comment (AC2)

**Dafina Kikaj, Senior Scientist**
**National Physical Laboratory**
**Teddington, UK**

**15ᵗʰ October 2024**

**Dr David Griffith, Editor**
**Atmospheric Measurement Techniques**

Dear Dr Griffith,

Please find below our response to the comments from the **anonymous referee #2** on our paper: **"Direct high-precision radon quantification for interpreting high frequency greenhouse gas measurements"**.

We would like to thank the reviewer for taking the time to look over our manuscript and provide us with comprehensive constructive feedback.

Best regards,

Dafina Kikaj
(on behalf of all co-authors)

Summary

Kikaj et al. present a technically-oriented paper on the best practices to create reliable long-term timeseries of atmospheric radon from a two-filter system. This study also includes recommendations and information for atmospheric radon measurements in general as well as a comparison of radon measurements to high-resolution greenhouse gas data. The title does not fully reflect the detailed technical nature of the paper in my opinion.

General comments

Overall, the paper is well-written, clearly structured and easy to follow. The authors chose great figures to illustrate and communicate their findings and recommendations. The literature cited does however seem a bit arbitrary as there are more studies using radon or the Radon-Tracer-Method. However, as this is not a review paper this is only a minor issue which can be fixed by highlighting that the reference chosen are only a subset of relevant studies. Despite the focus on radon and the technical nature of the manuscript it will definitely be of interest to experts in the field and other readers of AMT, thus I recommend publication after some minor and technical issues have been addressed.

**Thank you very much for your positive evaluation of our manuscript. We have carefully addressed your specific comments in the sections below. We appreciate your time and feedback.**

Specific and technical comments

P2 L 25: It would seem prudent to highlight (make the distinction) that the uncertainty of GHG emissions is mostly an issue for non-CO2 GHGs or carbon cycle feedbacks, while fossil fuel CO2 emissions are typically much better constrained, especially in countries reporting under UNFCCC Annex I

**Thanks.** *The following change has been made in the revised text:*

*"Since there can be large uncertainties associated with the spatial and temporal variability of these emission factors across sectors, as well as potential biases from unaccounted sources, especially for non-CO2 GHGs, it is prudent to seek independent verification of the resulting emission estimates."*

P2 L45: Here and elsewhere: Suggest to add "for example" before citations to highlight that this is far from a comprehensive list.

A quick googles-scholar search find many more relevant studies on both the use of radon as a transport modelling tracer (e.g. https://doi.org/10.1016/j.jenvrad.2004.03.033, https://doi.org/10.3402/tellusb.v65i0.18681, https://doi.org/10.5194/acp-15-1175-2015) or the use of atmospheric radon and GHG data for the radon tracer method (e.g. https://doi.org/10.5194/acp-7-3737-2007, https://doi.org/10.5194/acp-21-17907-2021, https://doi.org/10.3402/tellusb.v65i0.18037, https://doi.org/10.1080/1943815X.2012.691884).

**Thank you. I have ensured that "e.g." is consistently used throughout the manuscript when the citations provided are not a comprehensive list, and I have also included the suggested references.**

P3 L53: also many more studies than just Tolk et al.
**We have added "..here errors are related to the resolution of the model (e.g. Geels et al., 2007; Gerbig et al., 2008; Liu et al., 2011; Munassar et al., 2023; Tolk et al., 2008).**

P3 L58: Worthwhile to cite some papers that use radon as ATM performance tracer.
**We have added (e.g. Baker et al., 2006; Dentener et al., 1999; Tolk et al., 2008; Zhang et al., 2021)**

P3 L69: What is meant by local scale? Urban scale or even facility scale? Yver-Kwok used RTM to estimate emissions from a waste water treatment plant - this was the most 'local' study I was able to find: https://doi.org/10.5194/amtd-6-9181-2013
**In reality, our discussion has not focused on RTM but rather on atmospheric processes. We apologize for any confusion. By "local scale," we are referring to mesoscale processes.**
*The following change has been made in the revised text:*
**"It is therefore considered a powerful and convenient tracer at meso, synoptic and global scales for improving, testing and validating atmospheric models (Chambers et al., 2015, 2019b; Israël et al., 1966; Jacob et al., 1997; Taguchi et al., 2002; Zhang et al., 2021)."**

P9 L200 and elsewhere: here the flow is reported in liters per "m", but before minutes were abbreviated as min (P8 L193). Please make sure to use consistent abbreviations and units throughout the manuscript.
**This has now been corrected!**

P12 L341: please consider replacing 'networked' with a proper description.
*The following change has been made in the revised text:*
**"However, calibrations and backgrounds can also be remotely reconfigured and initiated if the computer has network connectivity."**

Figure 3: here the internal flow is reported as m3 s-1, while later sections refer to the internal flow in m s-1 (which is odd). Please be sure there is consistent use of units throughout the manuscript.
**We acknowledge that we mistakenly presented the internal flow rate in $m^3\ s^{-1}$ in some instances (e.g., Figure 3). We recognize the confusion regarding the use of velocity to describe flow rate. In this case, the value "5.5 m/s" refers to the** linear flow rate **(velocity) rather than the volumetric flow rate.**

*The following change has been made in the revised text:*
**"The flow rate of the internal flow loop should be sufficient to exchange all the air within the detector through the measurement head in less than 3 minutes (the half-life of $^{218}Po$). Flow within the central pipe of the detector is measured as a velocity (V) with an insertion probe at the centre of the 50 mm ID pipe. Based on the typical velocity profile in a pipe, the actual flow rate is estimated as 80% of the maximum flow rate $Q_{max}$. While a flow rate of around 5.5 m $s^{-1}$ is technically sufficient for this purpose, a faster rate**

is desirable, and values of 6–12 m s⁻¹ are typically achievable (based on individual blower performance and flow impedance of the measurement head)."
**Addition:** ($Q_{max} = 6x10^4\ p\ (0.025^2)$ (L.min⁻¹)).

P14 361f: I assume flow rates are supposed to be "m3 s-1" not "m s-1" here
**Please see our response to your previous comment.**

P14 369: Pressures are given in Pa, while the figure above suggest that the instrument record hPa, why the unnecessary conversion and not put 1-1.2hPa here?
**Thanks. Done.**

Table 1: same unit issues sometimes per minute is m-1 then min-1 and then m is used for meters in the same table.
**We apologize for this omission. It has now been corrected throughout the manuscript.**

P18 L456: suggest to remove ("to arrive") to clarify the sentence.
**Corrected.**

P18 L469: Shouldn't this be "radon activity concentration"?
**The detector reports scintillation counts over a 30-minute interval. This is why we did not use the term "radon activity concentration" as suggested, since the reported values are based on counts rather than direct measurements of radon activity concentration.**

**P20 L511: Possion -> Poisson**
**Corrected.**

P23 L577: should be: "atmospheric trace gas constituents". The bulk gas concentrations (N2, O2, Ar) hardly change in the troposphere/
**Corrected.**

P27 L662: Is this new scientific information or is some of this information also in the manual for this ANSTO instrument, if so, it should be referenced.
**Some of this information is indeed included in the ANSTO commissioning report. We referenced to this manual in the revised manuscript.**

P27 L673: What is this mobile calibration standard transfer device and where can it be acquired or requested?
*The following change has been made in the revised text:*
**Information about the availability of these transfer standard instruments can be obtained by contacting radon@ansto.gov.au.**

P27 L684: Please provide information on the supplier/manufacturer of the Burkert calibration or cite a document that describes it use/function.
**Information about this can be obtained by contacting radon@ansto.gov.au. We also cited Chambers et al. 2022 (https://doi.org/10.5194/adgeo-57-63-2022).**

P28 L709: was is "production code"?

**The Rust-based implementation of the deconvolution algorithm produces results which are similar to the original Python-based code. Its advantages are mostly related to usability: (1) the installation size is < 10Mb vs 2.6Gb, (2) the user is presented with a well-defined command-line interface rather than having to write a Python script, (3) a Windows binary is available in addition to a Linux binary, compared with the Linux-only Python version, (4) it is a simple one-command installation, and (5) the Rust version executes more quickly (but this is not very important, it does not execute much more quickly).**

**Considering it now, the distinction made in this paragraph between the two version of the deconvolution code is not particularly helpful, and we will remove the mention of the two implementations of the deconvolution code so that**
*Original text:*
**"Regarding the deconvolution of near real-time data, a newly developed Rust-based "production code" should be employed. This code typically takes up to one minute to deconvolve a day's worth of data. However, there is a Python-based code "research code" available for the level 1 data product."**
*Will be changed to*
**"For near real-time measurements, deconvolution can be run routinely with the code taking about one minute to process one day's worth of data. Deconvolution is then re-run in postprocessing after the finalisation of the calibration coefficients and background count rate timeseries."**

P29 L721 and section before: This description is helpful, however, the reader is left hanging. Where is all of this data going? Is there a central repository or a global database users can access. If not, is this something you recommend to be created?

**Thank you for your feedback. While we did not propose a specific central repository or global database, we acknowledge that the establishment of such a system is a decision that should be carefully considered by relevant stakeholders.**
**For the UK radon network, we currently utilize the Centre for Environmental Data Analysis (CEDA) for data storage and access. Additionally, for the ICOS (Integrated Carbon Observation System) network in Europe, the ICOS portal could serve as a potential platform for radon data accessibility. Regarding the GAW/WMO sites the WDCGG could be used.**
**However, the choice of a data repository depends on various factors, including data management protocols, accessibility requirements, and stakeholder engagement. We believe it is important for users and decision-makers to explore these options further to determine the most suitable solution for data sharing and accessibility.**

P29 L724: This paper only tangentially talks about 'real-time' data. Most of the things discussed here are about high temporal resolution instead. Given the focus on calibrations and the clear week to 5 year time-scale of calibration it seems odd to mention real-time. The data is flagged monthly, calibrated quarterly and deconvolution of the data is done every 6 month (or recommended to be done on that schedule section 5.4), hence reliable data is only available with months delay, i.e. very far from real-time. Also, who would really need/want actually real-time radon data? Reporting data a few hours or even days delayed seems perfectly fine for virtually all applications.

**Thank you for your feedback. We would like to clarify that when we do not refer to "real-time" data, we refer "near real-time," which can be understood as data available on a daily, weekly, or monthly basis. As stated in the first paragraph of this subsection "Each temporal scale is associated with different relative uncertainties, which should be clearly documented in the metadata". Additionally, while calibration occurs quarterly and deconvolution is recommended every six months, we can utilize the most recent fitted background and calibration values, which is the primary purpose of developing this fitted algorithms. Our deconvolution code can process one day's worth of data in approximately one minute, allowing us to deliver data efficiently.**
**We believe that having the option for "near real-time" data can significantly enhance operational monitoring and decision-making processes, particularly for evaluating the integration of operational data models.**

Table 2A and general: again internal flow is reported in m s-1, but now external flow is L min-1 instead of L m-1 before...
**Corrected.**

---

## Author Comment (AC3)

**Dafina Kikaj, Senior Scientist**
**National Physical Laboratory**
**Teddington, UK**

**15th October 2024**

**Dr David Griffith, Editor**
**Atmospheric Measurement Techniques**

Dear Dr Griffith,

Please find below our response to your comments on our paper: **"Direct high-precision radon quantification for interpreting high frequency greenhouse gas measurements"**.

We would like to thank you for taking the time to look over our manuscript and provide us with comprehensive constructive feedback.

Best regards,

Dafina Kikaj
(on behalf of all co-authors)

With apologies for the long response time, 2 referees have now submitted reports which recommend acceptance in AMT, require no major revisions but request a number of valid minor and technical corrections or enhancements. I encourage the authors to incorporate responses to these comments and prepare a revised manuscript. No major revisions are required.

In addition I have some further, minor technical suggestions that could be included at this stage:

Abstract (and conclusion) - use of "precision" when in most case you mean "accuracy" or sometimes "accuracy and precision". Please revise this usage.
**Thank you. We acknowledge this distinction and have made the necessary revisions in the manuscript to ensure that "accuracy" and "precision" are used appropriately.**

L62: it would be useful here to specify 222Rn rather than just Rn to distinguish from other isotopes in this context
**Corrected!**

L200 and many instances thereafter: please use L min(-1) not L m(-1) for flows. In SI "m" is the symbol for metre. There also cases further on where you have correctly used L min(-1). Reviewers also noticed this, please check the MS for all instances and correct them.
**We apologise for this omit. It is corrected now throughout the whole manuscript.**

p14: Flows are given as m s(-1). Is this really a linear flow rate of metre/s? Flow rate is usually measured as vol/time, but I do not think you mean m(3) s(-1). Reviewers also noted this, please clarify.
**This is a linear flow rate. We have corrected this now.**

**L366 I was confused by your use of a micro flow meter to estimate over pressure in Pa. Can you clarify this, please?**

The main detector delay volume (and entire inlet line downstream of the Becker stack blower) is kept at positive pressure with respect to ambient conditions (usually between 100-200 Pa). This overpressure minimises the likelihood that ambient radon/thoron progeny at high ground-level concentrations will directly enter the detector in the event of a leak developing in the detector's delay volume or inlet line.

During normal operation the detector's overpressure is monitored using a micro mass airflow sensor located within the data logger box (via the ta port in the detector's bulkhead; Figure 9). At the time of commissioning a hand-held differential pressure sensor is usually used to relate flow rate reported by the micro mass flow sensor to a differential pressure between the tank and ambient atmosphere for a range of overpressure values (between 80 – 180 Pa).